TOOLS

# Recombinant biosensors for multiplex and super-resolution imaging of phosphoinositides

Hannes Maib[1], Petia Adarska[3], Robert Hunton[1], James H. Vines[1], David Strutt[1], Francesca Bottanelli[3], and David H. Murray[2]

**Phosphoinositides are a small family of phospholipids that act as signaling hubs and key regulators of cellular function. Detecting their subcellular distribution is crucial to gain insights into membrane organization and is commonly done by the overexpression of biosensors. However, this leads to cellular perturbations and is challenging in systems that cannot be transfected. Here, we present a toolkit for the reliable, fast, multiplex, and super-resolution detection of phosphoinositides in fixed cells and tissue, based on recombinant biosensors with self-labeling SNAP tags. These are highly specific and reliably visualize the subcellular distributions of phosphoinositides across scales, from 2D or 3D cell culture to *Drosophila* tissue. Further, these probes enable super-resolution approaches, and using STED microscopy, we reveal the nanoscale organization of PI(3)P on endosomes and PI(4)P on the Golgi. Finally, multiplex staining reveals an unexpected presence of PI(3,5)P$_2$-positive membranes in swollen lysosomes following PIKfyve inhibition. This approach enables the versatile, high-resolution visualization of multiple phosphoinositide species in an unprecedented manner.**

## Introduction

Membrane identity is key to cellular function. A crucial hallmark of this identity are the phosphoinositides, a small family of phospholipids generated by phosphorylation of their phosphatidylinositol (PI) headgroup. Through the stepwise addition and removal of phosphate groups by lipid kinases and phosphatases, eight distinct phosphoinositides can be generated that recruit specific effector proteins and thereby regulate a myriad of cellular functions. As such, the different phosphoinositides are involved in basically every cellular event involving membrane dynamics, ranging from cytokinesis to cell migration, autophagy, T-cell activation, and membrane contact site regulation, as well as in their most well-known role as regulators of membrane trafficking (Edwards-Hicks et al., 2023; Gulluni et al., 2021; Jiménez et al., 2000; Mesmin et al., 2013; Tremel et al., 2021) see (Posor et al., 2022 for recent review). In line with their cell biological importance, continuing efforts are being undertaken to establish and refine tools to visualize their subcellular distribution.

Key to the visualization of phosphoinositides is the identification of effector domains that specifically recognize the differentially phosphorylated headgroups (Hammond and Balla, 2015). These effector domains are often conserved Pleckstrin or Phox homology (PH and PX) domains but can also be structural folds such as GRAM, FYVE, and Zinc finger domains, or the WD40 propeller found in WIPI1 and 2 (Dooley et al., 2014). To

further complicate matters, some proteins bind to specific phosphoinositides through polybasic regions without any conserved secondary structure, such as Exoc7 (Exo70) as part of the exocyst complex (He et al., 2007; Maib and Murray, 2022). Through the combined effort of the community, these effector domains have been optimized to generate biosensors to detect phosphoinositides (Wills et al., 2018). With the recent addition of a PX domain from *Dictyostelium discoideum* that recognizes PI(3,5)P$_2$, (Vines et al., 2023), we now possess biosensors for each of the eight different phosphoinositides (albeit with uncertainties about PI(5)P). A common feature of many of these biosensors is the use of tandem and triple repeats to increase binding through an avidity effect, as is the case for the 3xPH domain of TAPP1, the 3xPHD domain from ING2, the 2xFYVE domain of Hrs, and the 2xPX domain of SnxA to recognize PI(3,4)P$_2$, PI(5)P, PI(3)P, and PI(3,5)P$_2$, respectively.

The most common way to visualize the subcellular localization of phosphoinositides is to ectopically overexpress the lipid effectors fused to fluorescent proteins as biosensors (Wills et al., 2018). This approach allows for the detection of membrane identity changes in live cells and has been of immense value over the last decades. However, it also suffers from a major drawback: as these biosensors are generated to have a high affinity, they can all potentially out-compete endogenous effector proteins and thereby perturb the very pathway that is under

[1]School of Biosciences, University of Sheffield, Sheffield, UK; [2]Division of Molecular, Cell and Developmental Biology, School of Life Sciences, University of Dundee, Dundee, UK; [3]Institut für Biochemie, Freie Universität Berlin, Berlin, Germany.

Correspondence to Hannes Maib: h.maib@sheffield.ac.uk.

investigation. In contrast, biosensors with lower affinity and expression levels are themselves out-competed by endogenous effectors. Furthermore, this overexpression-based approach is limited by the ability to deliver DNA into cells or requires time and labor-extensive genomic engineering. It is therefore still a challenge for experimentalists to visualize these critically important lipids, especially in more complex systems such as 3D cell cultures or whole tissues.

One way to circumvent these pitfalls has been to stain for phosphoinositides after fixation and permeabilization using immunocytochemical techniques based on recombinant biosensors (Gillooly et al., 2000; Watt et al., 2002) and antibodies (Hammond et al., 2009; Maekawa and Fairn, 2014; Marat et al., 2017). While this approach is unable to provide dynamic information, it does avoid perturbation of phosphoinositide signaling and does not require overexpression. However, the need for fixatives and detergents requires careful optimizations to avoid artifacts, and the available antibodies require different staining protocols for the visualization of distinct pools of their respective targets (Hammond et al., 2009). As such, there is currently no reliable tool to comprehensively visualize all subcellular phosphoinositide pools with a unifying staining approach.

To address these shortcomings, we have developed recombinant biosensors against all eight phosphoinositides in combination with self-labeling protein tags (SNAP). With the exception of the only known biosensor against PI(5)P, all of these probes are easy to purify using standard bacterial expression systems and demonstrate excellent specificities, as determined using an in vitro supported lipid bilayer approach. Using a single, unified staining protocol, we show that these probes reliably visualize at least six out of the eight phosphoinositide species after fixation and permeabilization. We successfully reproduce the known subcellular localizations of these phosphoinositides and verify the specificity of the staining approach using a range of lipid kinase inhibitors. With the exception of PI(5)P, and with some caveats in detecting PI, this toolkit enables the reliable and reproducible detection of PI(3)P, PI(4)P, PI(3,4)$_2$, PI(3,5)$_2$, PI(4,5)$_2$, and PI(3,4,5)$_3$.

Using this toolkit, we elegantly demonstrate that endocytic cargo transitions from early PI(3)P-positive endosomes to late PI(4)P-positive compartments. The ease of conjugating distinct fluorophores to the SNAP tag enables the use of inorganic dyes, including those that are compatible with super-resolution STED microscopy. This reveals the nanoscale organization of PI(3)P at endosomes and of PI(4)P at the Golgi while furthermore verifying the preservation of subcellular membranes. Labeling with distinct fluorescent dyes in vitro enables multiplex imaging and allows for the detection of phosphoinositides across scales and model systems as shown by staining in HeLa cells, NMuMG spheroids, and *Drosophila* pupal wings. To highlight the versatility of this approach, we investigated the interdependence of phosphoinositide conversion by multiplex staining for PI(3)P, PI(4)P, and PI(3,5)$_2$ after inhibition of lipid kinases in MIA-Paca2 cells. These experiments unveil an unexpected sequestration of residual PI(3,5)$_2$ positive membranes in swollen lysosomes following PIKfyve inhibition.

## Results

### Recombinant biosensors are straightforward to produce and are highly phosphoinositide specific

The cell biology community has spent considerable effort to identify effector domains of proteins that recognize specific phosphoinositides for use as biosensors (see table in Fig. 1). To test their suitability as recombinant probes, we designed effectors with an N-terminal 6xHis tag followed by a SNAP tag and a flexible linker region for bacterial expression. With the exception of the 3xPHD domain of ING2 (see Discussion), these recombinant biosensors are readily purified with high yields from *E. coli* using a standard three-step purification procedure of Ni$^{2+}$ affinity followed by ion exchange and size-exclusion chromatography. The inclusion of a SNAP tag then allows for versatile in vitro labeling of the probes using inorganic fluorophores for multiplexed and/or super-resolution microscopy (Fig. 1).

To test the specificities toward their respective targets, we labeled the biosensors with SNAP-Surface Alexa Fluor 488 in vitro and formed supported lipid bilayers on silica beads with liposomes composed of 95% 1-palmitoyl-2-oleoyl-glycero-3-phosphocholine (POPC) and 5% of each phosphoinositide. These membrane-coated beads are an ideal substrate to evaluate binding specificities, as they faithfully mimic many properties of biological membranes (Pucadyil and Schmid, 2010) and enable quantification of effector binding to thousands of individual beads. The quality of the lipid bilayers was verified by including 0.1% of fluorescently labeled lipids (here Atto647N-DOPE) and using this signal to generate a membrane mask to quantify the recruitment of the biosensors (Fig. 2).

The biosensor for PI is a modified version of a bacterial, PI-specific PLC (Pemberton et al., 2020) and shows good specificity toward PI albeit with lower affinity compared with the other probes (Fig. 2 a). The biosensor for PI(3)P is a well-known tandem repeat of the FYVE domain from HRS (Burd and Emr, 1998) and shows excellent specificity and affinity for PI(3)P with some weak binding toward PI(3,5)$_2$ (Fig. 2 b). The probe for PI(4)P is the effector domain of SidC (also known as P4C) (Dolinsky et al., 2014) and shows strong binding to PI(4)P with some background binding to the other monophosphates, PI(3)P and PI(5)P (Fig. 2 c). The probe for PI(5)P is the triple repeat of the PHD domain from ING2 (Gozani et al., 2003) and shows little specificity. Even though this probe shows stronger binding to PI(5)P compared with PI, PI(3)P, PI(3,4)$_2$, and PI(4)P, it binds even more strongly to the other 5′ containing di- and triphosphates, suggesting a charge-based binding effect in addition to 5′ binding (Fig. 2 d). Therefore, even though this biosensor displayed promising results from overexpression in cells, it is unlikely to be suitable as a recombinant biosensor. The biosensor for PI(3,4)$_2$ is the triple repeat of the C-terminal PH domain from TAPP1 (Goulden et al., 2019) and shows good specificity and affinity with some binding to the di- and triphosphates (Fig. 2 e), likely due to the strong positive charge of this probe (with a pI of 9.7). The biosensor for PI(3,5)$_2$ is the recently described tandem repeat of a PX domain from *D. discoideum* (Vines et al., 2023) and shows excellent specificity and affinity with almost undetectable binding to any of the other phosphoinositides (Fig. 2 f). The biosensor for PI(4,5)$_2$ is a well-

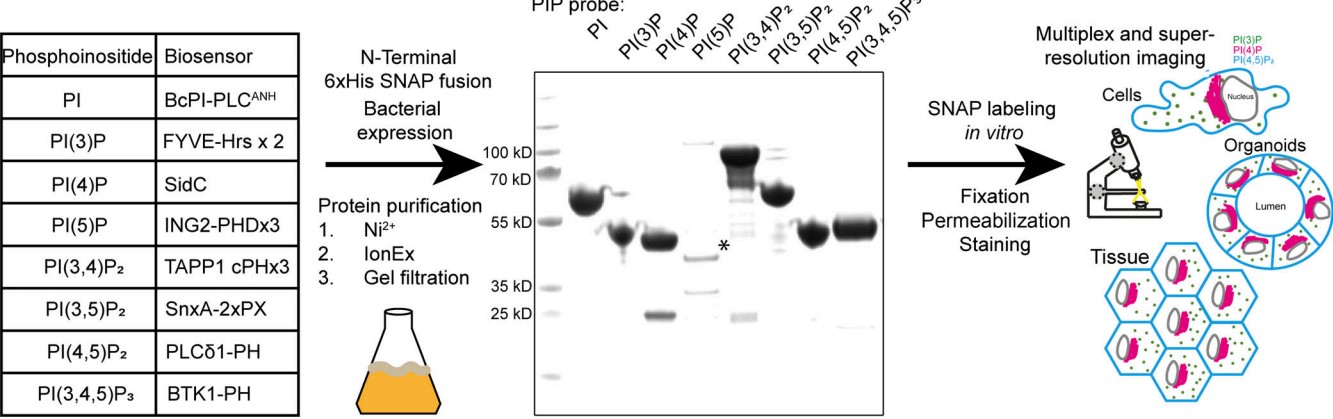

Figure 1. **Scheme for the generation of recombinant biosensors for detection of phosphoinositides in fixed and permeabilized cells.** Left: Table of biosensors used for the detection of each phosphoinositide. Middle: Coomassie stained gel of purified probes after a three-step purification from *E. coli*. The asterisk denotes the soluble fraction of the PI(5)P biosensor, which is only stable at low concentrations. Source data are available for this figure: SourceData F1.

established PH domain from PLC-δ1 (Garcia et al., 1995) and shows reassuring specificity (Fig. 2 g). And finally, the biosensor for PI(3,4,5)P₃ is the PH domain from BTK (Fukuda et al., 1996; Salim et al., 1996) and shows good specificity with some minor binding to PI(3,4)P₂ (Fig. 2 h). These comprehensive and directly comparable in vitro data validate the specificity for each of the recombinant biosensors and suggest their suitability for use in staining approaches.

## Subcellular visualization of nearly all phosphoinositide species by a unified staining approach

Visualization of phosphoinositides using these recombinant biosensors requires cell fixation and permeabilization. Importantly, multiplex imaging requires a unifying staining protocol that is easy to use while preserving all of the distinct phosphoinositide pools, whether they are at the plasma membrane, Golgi, or endosomes. To achieve this objective, HeLa cells were fixed with prewarmed (37°C) 4% PFA with 0.2% glutaraldehyde and permeabilized using 0.5% Saponin on ice and stained with 500 nM of each of the separate biosensors labeled with Alexa488 (Fig. 3) (see Materials and methods for more details). To verify that this approach does not alter membrane organization, we ectopically overexpressed the PI(4)P probe tagged with eGFP and imaged the exact same cells before and after staining with the recombinant biosensor labeled with Alexa647 (Fig. S1). Reassuringly, membrane organization seems unaltered following this staining approach. The recombinant biosensor detects the same lipid pools at the plasma membrane and subcellular compartments as in cells expressing the biosensor at low levels. However, at high expression levels, PI(4)P is fully occupied and the recombinant probe is unable to detect it (Fig. S1). Therefore, the recombinant probe detects the same target as the overexpressed biosensor, and at high expression levels, all available phosphoinositides are fully occupied.

Using this protocol, the staining for PI in HeLa cells shows a rather diffuse pattern with some membrane staining but poor contrast (Fig. 3 a). This is likely due to the relatively low affinity of this probe and the almost ubiquitous presence of this

phosphoinositide on subcellular membranes. The staining for PI(3)P on the other hand shows excellent signal-to-noise ratio with clear labeling of endo/lysosomal structures throughout the cell (Fig. 3 b). The PI(4)P staining shows excellent labeling of all known pools of this phosphoinositide, including the Golgi, trafficking vesicles, and the plasma membrane, especially at junctions between cells (Fig. 3 c). The cellular staining using the recombinant biosensor against PI(5)P shows no localization to membranes, with the only clear signal in the nucleolus (Fig. 3 d). This is most likely due to the strong negative charge of open DNA and agrees with the mainly unspecific, charge-based binding seen with supported lipid bilayers (Fig. 2 d). Staining with the PI(3,5)P₂ biosensors reveals faint vesicular staining, in agreement with the low steady-state levels of this rare phosphoinositide in HeLa cells (Fig. 3 e). In agreement with its known localization, the PI(3,4)P₂ biosensor stains weakly the plasma membrane with enrichment at membrane ruffles (Fig. 3 f). The PI(4,5)P₂ biosensors reveal a strong plasma membrane and a weak intracellular staining, in agreement with the well-described presence of minor pools of this phosphoinositide on intracellular membranes (Fig. 3 g). Finally, the biosensor for PI(3,4,5)P₃ shows selective staining of subregions of the plasma membrane such as the leading edge and structures reminiscent of macropinocytic cups (Fig. 3 h).

To verify the specificities of each of these stainings, we carried out extensive characterizations using commercially available lipid kinase inhibitors (Figs. S2 and 6). The diffuse staining against PI was the most challenging to address. Since this lipid is the substrate for PI3-, PI4-, and PI5-Kinases, we reasoned that by blocking their activity using an inhibitor cocktail, we would detect an increased level of PI on some membranes. Indeed, following this treatment, we could detect an increased signal of PI at the membrane of vacuolar structures (Fig. S2 a). However, this staining still shows a high amount of background signal, making the confident detection of PI challenging. The endo/lysosomal staining of PI(3)P was completely abolished by treatment with the pan PI3K inhibitor Wortmannin (Fig. S2 a and Fig. 6). PI(4)P is reliably detected at the plasma membrane,

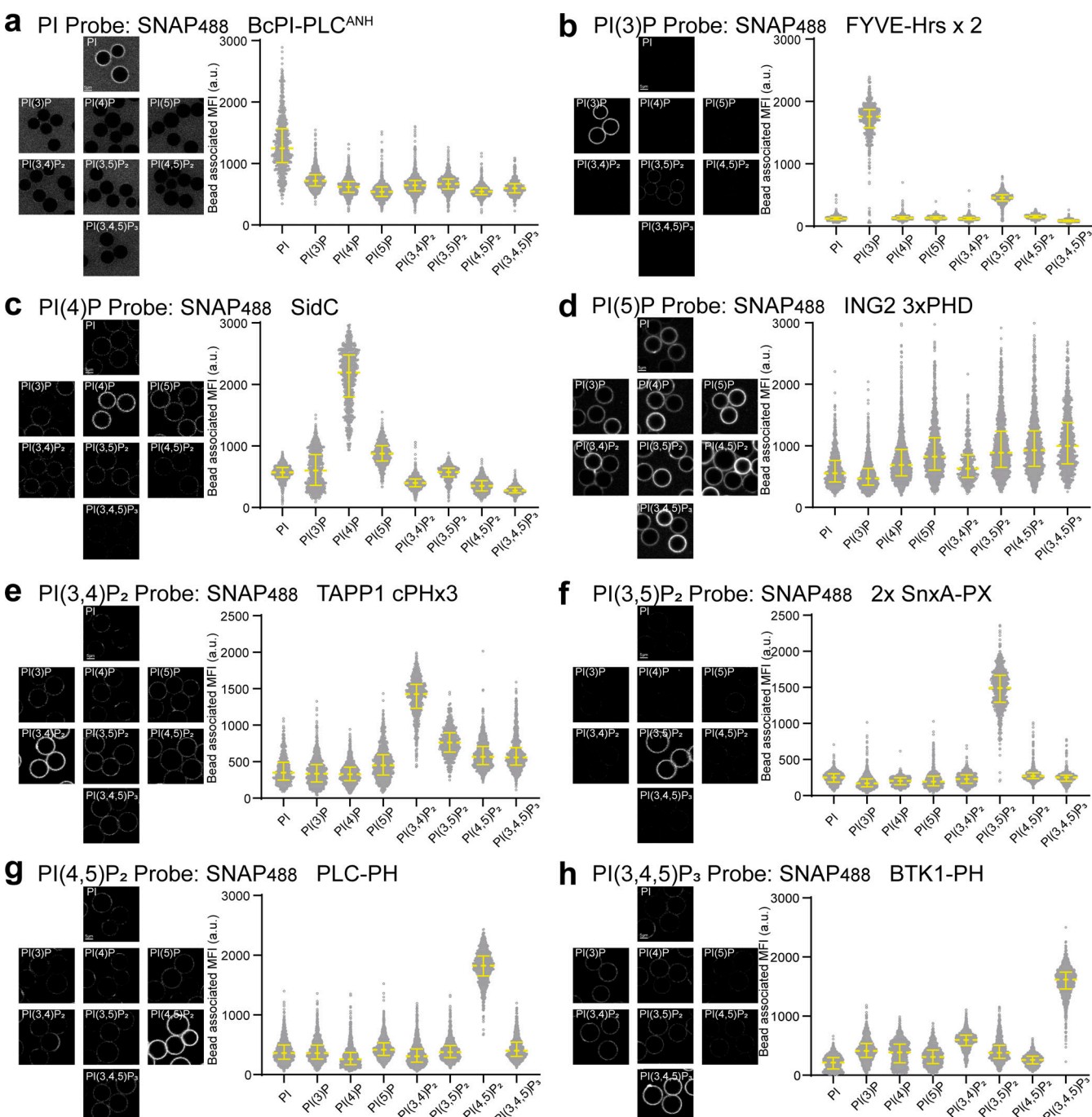

Figure 2. **In vitro verification of recombinant biosensors. (a–h)** the indicated biosensors were labeled with Alexa488 and added at 200 nM final concentration to membrane-coated beads containing 5% of the indicated phosphoinositide. Mean fluorescence intensity (MFI) of the biosensor bound to each individual bead was determined by semiautomated segmentation using the fluorescent DOPE lipid as a mask in ImageJ. Data are presented as a median with an interquartile range from $n$ = 473–1,642 individual beads pooled from three technical repeats for each phosphoinositide and each biosensor.

endosomes, and the Golgi, and treatment with kinase inhibitors against the different PI4K isoforms leads to the selective depletion of these pools (Fig. S2 b). To test the specificities of $PI(4,5)P_2$, $PI(3,4)P_2$, and $PI(3,4,5)P_3$, we used EGF stimulation to induce membrane ruffles in which these phosphoinositides were enriched (Araki et al., 2007; Kotani et al., 1994; Schink et al., 2016). Treatment with the PI3K inhibitor LY294002 depleted $PI(3,4)P_2$ and $PI(3,4,5)P_3$ in these ruffles while not

affecting $PI(4,5)P_2$ (Fig. S2 c), in agreement with $PI(3,4)P_2$ being derived from $PI(3,4,5)P_3$ (Goulden et al., 2019). The specificity of $PI(3,5)P_2$ staining is addressed in Fig. 6. Finally, by using a one-step His purification followed by removing the tag using PreScission protease, we verify that the presence of the 6xHis tag does not affect the staining and enables the use of a simplified one-step purification, if so desired (Fig. S3). In summary, this approach demonstrates a unifying and well-characterized

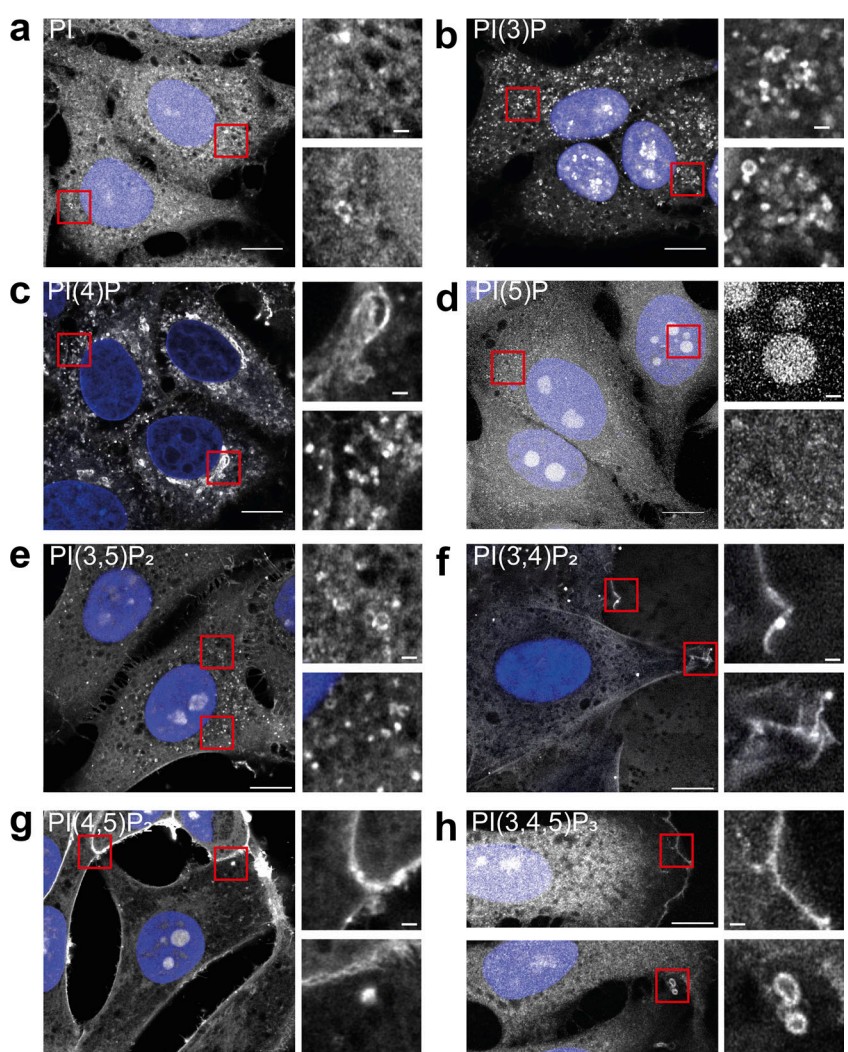

Figure 3. **Cellular staining of phosphoinositides in fixed cells using recombinant biosensors. (a–h)** HeLa cells were grown on coverslips and fixed and permeabilized as described in the Materials and methods section. Phosphoinositides were detected using recombinant biosensors conjugated to Alexa488 at 500 nM final concentration and imaged using Airyscan microscopy. DAPI was included to visualize the nucleus and show it in blue. Scale bar = 10 and 1 μm in inserts.

protocol for staining and visualization of nearly all phosphoinositide species in fixed and permeabilized cells.

## Recombinant biosensors in combination with STED microscopy reveal the nanoscale organization of phosphoinositides

PI(3)P and PI(4)P are the most abundant intracellular phosphoinositide and both localize to endosomes. To gain mechanistic insights into the function of these signaling lipids on distinct endosomal populations, we performed a pulse-chase assay using fluorescently labeled transferrin and multiplex stained against PI(3)P and PI(4)P after varying times of internalization. The transferrin receptor is well known to be constitutively trafficked through clathrin-mediated endocytosis, before being recycled back to the plasma membrane, and is often used as a housekeeping cargo (Maib et al., 2018). In agreement with previous work, we find that transferrin localizes to PI(3)P-positive endosomes at early timepoints (peak at 5 min), while increasingly localizing to PI(4)P-positive structures at later time points (10–20 min) post internalization (Fig. 4 a). This result is consistent with the membrane identity of endosomes being converted from PI(3)P into PI(4)P to mediate receptor recycling (Ketel et al., 2016).

To gain a better understanding of this sorting, we harnessed super-resolution microscopy of PI(3)P-positive endosomes. The use of SNAP allows for the incorporation of bright, photostable, inorganic fluorophores, such as the JaneliaFluor dyes (Grimm et al., 2020, 2021), that are exceptionally well-suited for stimulated emission depletion (STED) microscopy. Staining HeLa cells with the PI(3)P biosensor conjugated to JFX650 shows the well-established endosomal localization of this phosphoinositide in "microdomains" (Gillooly et al., 2003). When viewed using STED microscopy, however, it is revealed that these domains are indeed nanoscale membrane tubulations as well as smaller vesicles that cluster around these larger endosomes and potentially fuse with, or bud from them (Fig. 4 b), as has been seen previously using correlative light and electron microscopy (Franke et al., 2019). This highlights the nature of PI(3)P-positive endosomes as sorting hubs of internalized cargoes.

Understanding the nanoscale organization of phosphoinositides is key toward gaining novel insights into their functions. We therefore harnessed the strength of our staining approach in combination with STED microscopy to gain more insights into the organization of PI(4)P at the Golgi. HeLa cells expressing endogenously tagged Arf1[EN]-Halo were first stained live against

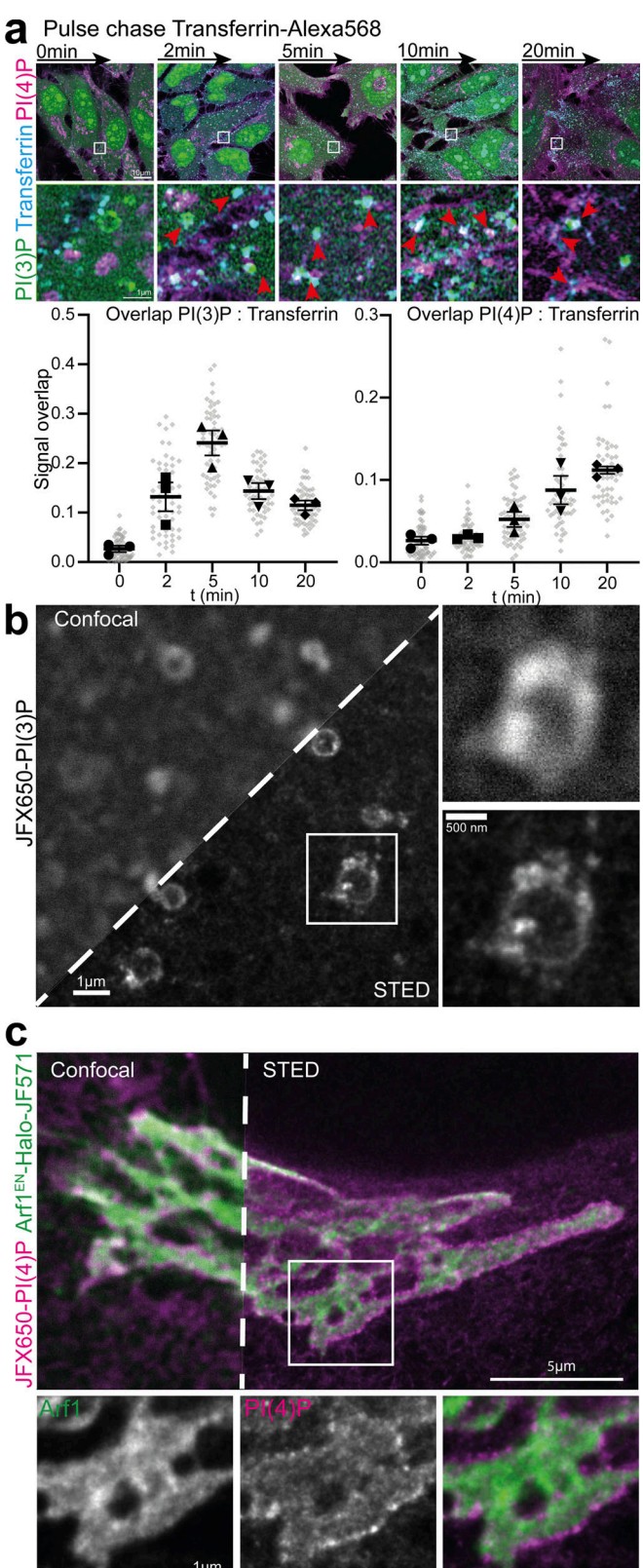

the Halo tag using the JF571 fluorophore before being fixed, permeabilized, and stained against PI(4)P using the recombinant biosensor conjugated to JFX650. Arf1 is a crucial factor in the generation of PI(4)P at the Golgi through recruitment of PI4K IIIβ (Highland and Fromme, 2021), and this combination of dyes allows for two-color STED imaging (Wong-Dilworth et al., 2023). Excitingly, the staining for PI(4)P reveals distinct nanoclusters of this lipid at the curved rims of the cisternae where budding of trafficking intermediates is thought to occur as well as diffuse staining throughout the Golgi (Fig. 4 c). The localization of the PI(4)P sensor is in fact reminiscent of COPI clusters (Wong-Dilworth et al., 2023). Thereby, this robust approach for the visualization of phosphoinositides enables super-resolution interrogation of membrane organization while preserving the structure of subcellular membranes.

## Multiplex detection of phosphoinositides across scales and model systems

A staining protocol, which preserves the subcellular organization of membranes and is compatible with the detection of each of the distinct phosphoinositides, opens up the possibility for multiplex imaging. As proof of principle, HeLa cells were stained with recombinant biosensors against PI(3)P, PI(4)P, and PI(4,5)$P_2$ conjugated to Alexa488, 546, or 647, respectively, and imaged using Airyscan microscopy (Fig. 5 a). Amongst numerous observations, we noted PI(4)P-positive tubules in direct contact with PI(3)P-positive endosomes in the vicinity of the PI(4,5)$P_2$-positive plasma membrane (Fig. 5 a [insert]). This multiplex approach highlights the intricate interactions of membranes with distinct phosphoinositide identities, critically important to the organization of organelle contact sites (Posor et al., 2022).

An important advantage of recombinant biosensors is that it avoids the need for overexpression. As such, it allows for the detection of phosphoinositides in cells and tissues that cannot easily be transfected and negates the need for genome engineering. To highlight this advantage, we grew NMuMG spheroids in Matrigel for 5 days and used the same staining protocol as previously to detect PI(3)P, PI(4)P, and PI(4,5)$P_2$ (Fig. 5 b). As in 2D cell culture, these three phosphoinositides show an intricate interaction at the subcellular level with strong labeling of PI(4,5)$P_2$ at cell–cell junctions. To further test the utility of this approach in whole tissues, we stained *Drosophila* pupal wings against PI(4)P and PI(4,5)$P_2$ in combination with conventional Phalloidin to visualize the actin cytoskeleton and cellular junctions. Consistent with the results from 3D culture, the recombinant biosensor against PI(4,5)$P_2$ shows strong labeling of cellular junctions throughout the whole tissue. Noticeably, PI(4)P is also enriched at these junctions albeit with a more diffuse staining pattern, indicating the presence of intracellular PI(4)P membranes in close proximity to the plasma membrane (Fig. 5 c). These observations are in good agreement with the

Figure 4. **Super-resolution STED imaging of nanoscale phosphoinositide organization. (a)** Pulse-chase assay in HeLa cells using fluorescently labeled transferrin followed by staining against PI(3)P and PI(4)P using recombinant biosensors. Signal overlap was determined using the ImageJ plugin Squassh. Data are mean ± SEM from three independent repeats with 16–20 cells each. **(b)** Endosome staining of HeLa cells using biosensor against PI(3)P conjugated with JFX650 as seen by confocal and STED microscopy. **(c)** Two-color STED microscopy of Golgi in HeLa cells endogenously tagged Arf1[EN]-Halo stained with JF571 and PI(4)P using recombinant biosensor conjugated to JFX650.

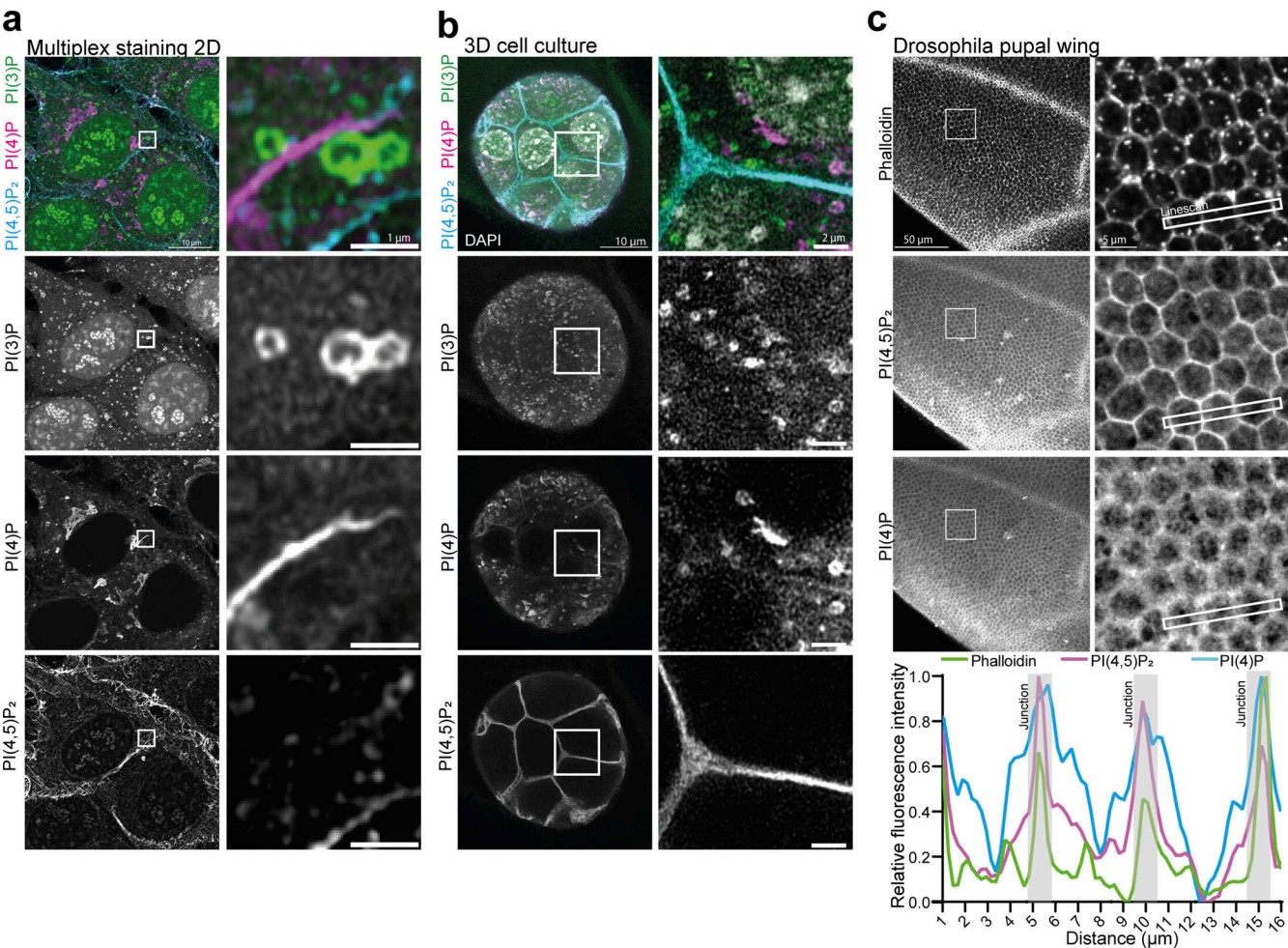

Figure 5. **Multiplex staining of phosphoinositides across scales. (a)** HeLa cells were fixed, permeabilized, and stained using recombinant biosensors against PI(3)P, PI(4)P, and PI(4,5)P$_2$, conjugated respectively to Alexa488, 546, and 647. **(b)** NMuMG spheroids were grown in Matrigel and stained with the same combination. **(c)** *Drosophila* pupal wings were dissected and stained with the PI(4,5)P$_2$ and PI(4)P biosensors conjugated to Alexa647 and 546, together with Phalloidin conjugated to Alexa488, to visualize the actin cytoskeleton and cellular junctions.

dynamic regulation of these two lipids at cell junctions in *Drosophila* tissue to drive the recruitment of polarity proteins through electrostatic interactions (Dong et al., 2015; Lu et al., 2022).

These experiments demonstrate conclusively that this toolkit enables the visualization of several phosphoinositide species simultaneously without the adverse effects of overexpression. This will enable the visualization of these crucial lipids across scales and model systems, from 2D to 3D cell culture and thin tissues.

### Staining reveals hidden pools of PI(3,5)P$_2$ following PIKfyve inhibition

Multiplex staining of phosphoinositides enables investigation of conversion cascades in a straight forward manner and avoids overexpression of multiple biosensors. To highlight this advantage and to make use of a novel PI(3,5)P$_2$ probe (Vines et al., 2023), we decided to focus on the conversion cascade leading to the generation of this rare phosphoinositide. Staining for PI(4)P, PI(3)P, and PI(3,5)P$_2$ in combination with Airyscan microscopy

reveals the close relationship of these three phosphoinositides (Fig. 6 a). While PI(4)P membranes are often in close proximity to PI(3)P-containing endosomes, it is rarely found on the same vesicle. PI(3,5)P$_2$, in contrast, is located on the same membrane as PI(3)P and enriched in domains that likely represent membrane tubulations and budding/fusion of smaller vesicles (Fig. 6 a). To investigate the conversion cascade of PI → PI(3)P → PI(3,5)P$_2$, we used specific kinase inhibitors to block each conversion step in MIA–Paca2 cells due to a low steady-state level of PI(3,5)P$_2$ in HeLa cells. As expected, treatment with the potent pan PI3K inhibitor Wortmannin (at 300 nM for 60 min) eliminates vesicular staining for PI(3)P as well as for PI(3,5)P$_2$, with minor effects on PI(4)P distribution (Fig. 6 b middle). Treatment with Apilimod, a highly selective and effective PIKfyve inhibitor (Cai et al., 2013) (at 1 µM for 60 min), leads to the accumulation of PI(3)P on endosomes as well as the formation of swollen lysosomes and the depletion of PI(3,5)P$_2$. We also observed some morphological changes in the organization of PI(4)P membranes (Fig. 6 b bottom). This confirms the specificity of the staining approach and that the generation of PI(3,5)P$_2$ is strictly

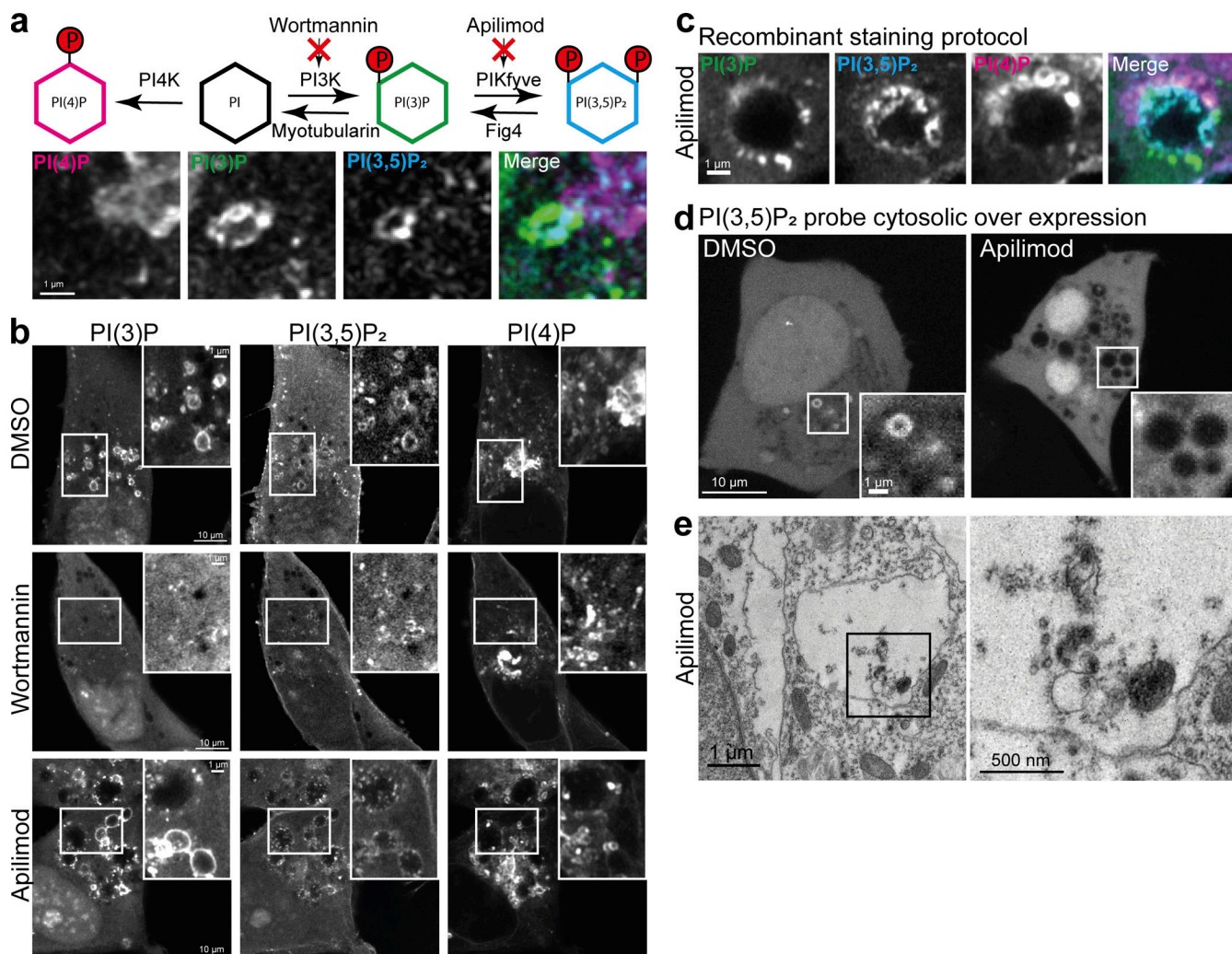

Figure 6. **Phosphoinositide interconversion and hidden PI(3,5)P₂ positive intraluminal membranes following PIKfyve inhibition. (a)** Interconversion scheme and multiplex staining of PI(4)P, PI(3)P, and PI(3,5)P₂. **(b)** MIA-Paca2 cells were treated with DMSO, 300 nM Wortmaninn, or 1 µM Apilimod for 60 min before fixation, permeabilization, and multiplex staining. **(c)** Example of swollen lysosomes following Apilimod treatment and stained against PI(4)P, PI(3)P, and PI(3,5)P₂. **(d)** PI(3,5)P₂ biosensor, fused to eGFP, was cytosolically overexpressed and cells were imaged life after Apilimod treatment. **(e)** Thin section electron micrograph of swollen lysosomes in MIA-Paca2 cells after treatment with Apilimod is shown.

dependent on the prior formation of PI(3)P, which accumulates on endosomes when its conversion into PI(3,5)P₂ is blocked.

Upon closer inspection of the images, we noticed that the PI(3,5)P₂ biosensor detects a distinct signal in the lumen of the swollen lysosomes upon PIKfyve inhibition (Fig. 6 c). Importantly, these intraluminal membranes are inaccessible to a cytosolically expressed PI(3,5)P₂ biosensor—highlighting an additional limitation of this conventional approach (Fig. 6 d). To address the nature of this staining, we investigated these swollen lysosomes using standard thin-section electron microscopy. This orthogonal technique confirms the presence of these intraluminal membranes, corresponding to the staining observed using the recombinant PI(3,5)P₂ biosensor. This unexpected detection of residual PI(3,5)P₂ membranes following PIKfyve inhibition highlights another advantage of the staining approach by making intraluminal membranes accessible that are otherwise hidden from the cytosolic overexpression of these probes.

## Discussion

The subcellular detection of phosphoinositides using fluorescence microscopy has been instrumental in the investigation of these signaling lipids. Through the combined efforts of the community, specific effector domains have been identified to detect each phosphoinositide and are widely used in overexpression-based approaches. This approach, however, often perturbs phosphoinositide signaling, is not possible in cell types that cannot be transfected, and is perilous for multiplex detection due to the need to express multiple probes. To address these shortcomings, we have generated a toolkit of recombinant biosensors for the detection of phosphoinositides for multiplex and super-resolution imaging in fixed and permeabilized cells.

These recombinant biosensors are easily purified by standard means and show excellent specificities in vitro, and the use of the SNAP tag allows for high versatility in the choice of inorganic fluorophores. These fluorophores benefit from vastly improved brightness and photostability compared with fluorescent

proteins and allow for the detection of minor phosphoinositide pools. Importantly, the use of a unifying fixation and permeabilization protocol for the detection of distinct subcellular pools allows for multiplex staining. This protocol is adapted from Hammond et al. (2009), where it was used for immunocytochemical detection of PI(4)P and PI(4,5)P$_2$ using commercial antibodies. However, these antibodies detect different phosphoinositide pools depending on the staining protocol. As such, after permeabilization with saponin, they detect PI(4)P at the plasma membrane but not at the Golgi. Conversely, permeabilization using Digitonin enabled the visualization of PI(4)P at the Golgi but abolished staining of PI(4)P and PI(4,5)P$_2$ at the plasma membrane (Hammond et al., 2009). Importantly, the recombinant biosensors that we have presented here allow visualization of all of these distinct phosphoinositide pools with the same saponin-based, staining protocol. Still, to reach all subcellular compartments, their surrounding membranes have to be disrupted using detergents, which might introduce artifacts. To address this concern, we directly observed the membrane organization of cells prior to, and after, our staining protocol, as well as using super-resolution STED microscopy. In each approach, we could confirm that subcellular organelles retain their structure. Therefore, these recombinant biosensors offer a clear advance for the subcellular detection of phosphoinositides.

However, it is also important to clearly outline the limitations of this approach. Perhaps most importantly, phosphoinositides will be partially occupied by effector proteins which occlude their detection to a certain extent, as we have directly demonstrated by high overexpression of fluorescently labeled effectors. Thus (as with genetically encoded biosensors), this approach is only able to detect free phosphoinositides, and the absence of evidence must not be confused with the evidence of absence. Minor phosphoinositide pools might always escape detection due to being highly occupied by endogenous effectors. Furthermore, when used for multiplex staining, the distinct spectral properties of various fluorophores means that their signal cannot be directly used to measure their relative quantities without careful prior calibrations. Finally, we caution against the overinterpretation of the nuclear staining using most of these biosensors. The probes against PI, PI(3)P, PI(5)P, PI(3,5)P$_2$, PI(4,5)P$_2$, and PI(3,4,5)P$_3$ all show nuclear staining patterns, and we believe that this is most likely unspecific due to the negative charge and the high availability of phosphate groups in open DNA. Importantly, the probes against PI(3)P, PI(3,5)P$_2$, and PI(3,4,5)P$_3$ show strong nuclear localization, which is completely unaffected by treatment with well-characterized lipid kinase inhibitors. While the role of nuclear phosphoinositides is an important field of research (Shah et al., 2013), it is unlikely that this staining approach is able to reliably detect them.

Out of this toolkit of biosensors, the probe to detect PI(5)P is the only one that is challenging to produce recombinantly. The only established probe to detect this elusive phosphoinositide is based on the zinc finger domain of ING2 (Gozani et al., 2003) and has shown promising results in overexpression-based approaches (Pendaries et al., 2006; Vicinanza et al., 2015). However, it is highly unstable as a recombinant protein and requires supplementation of Zn$^{2+}$ and aggregates at high concentrations, although it can be separated from its aggregates in low concentrations using size-exclusion chromatography. Disappointingly though, even after careful biochemical purification, the recombinant probe shows little specificity toward PI(5)P in vitro and only shows unspecific staining of the nucleolus when used to stain cells. Therefore, we do not recommend the use of this probe as a recombinant biosensor and have not included it in the Addgene deposition. Further work will have to be carried out to identify a reliable probe that works as a recombinant biosensor to detect PI(5)P.

Using this toolkit enables several advances in phosphoinositide biology. Especially, their nanoscale organization is crucial for the recruitment of effector proteins and the downstream regulation of cellular function. To shed more light on this, we made use of the development of inorganic fluorophores in combination with super-resolution STED microscopy and stained cells with recombinant biosensors against PI(3)P and PI(4)P. The organization of PI(3)P in microdomains on endo/lysosomes has been well described for over two decades (Gillooly et al., 2003); however, their precise organization has been somewhat unclear. By using STED, we could show that these domains are indeed nano tubulations and smaller vesicles in agreement with correlative light and electron microscopy of small GTPases on early endosomes (Franke et al., 2019). This nanoscale organization is supported by the role of PI(3)P as an effector for EEA1 in the tethering of early endosomes (Murray et al., 2016), as well as in the recruitment of ESCRT-0 in the formation of multivesicular bodies (Banjade et al., 2022) and in the regulation of the CCC complex in WASH-mediated recycling (Singla et al., 2019). Similarly, PI(4)P is a crucial identity determinant of the Golgi and regulates secretion (Waugh, 2019). The nanoclusters we have observed here are consistent with the clustering of this lipid through multiple effectors in the formation of AP1 and COPI coats (Lorente-Rodriguez and Barlowe, 2011; Tan and Brill, 2014) and opens up new avenues for future research. We believe that this approach of staining for phosphoinositides in combination with STED microscopy will reveal further insights into the nanoscale organization of phosphoinositides in the years to come.

The ability to reliably stain for phosphoinositides enables the investigation of these crucial lipids across scales and model systems. In addition to super-resolution microscopy, the versatility of the SNAP tag makes differential labeling and multiplex imaging possible. As proof of principle, we have performed triple staining of the most abundant phosphoinositides: PI(3)P, PI(4)P, and PI(4,5)P$_2$ in 2D as well as 3D cell culture. This highlights the close interactions between these membranes, as has been shown for the role of PI(4)P-containing membranes in regulating the scission of PI(3)P-positive endosomes (Boutry et al., 2023; Gong et al., 2021). To further highlight the versatility of our approach, we stained *Drosophila* pupal wing tissue against PI(4,5)P$_2$ and PI(4)P in combination with Phalloidin to visualize F-actin. This approach clearly detects high levels of both PI(4)P and PI(4,5)P$_2$ at cellular junctions throughout the whole tissue. We noticed that the levels of PI(4)P at junctions seemed to increase from 2D to 3D cell culture and tissue. While warranting further research, this increase in negatively charged

phospholipids in polarized systems is in good agreement with the recruitment of polarity proteins through electrostatic interactions (Dong et al., 2015; Lu et al., 2022). Importantly, the ability to visualize phosphoinositides in model systems that are difficult to genetically modify has the potential to reveal novel insights into their functions on the tissue scale and furthermore opens up the possibility to investigate these conserved lipids in eukaryotes across the evolutionary tree.

Multiplex staining also enables the investigation of phosphoinositide conversion cascades, as we have shown for the generation of PI(3,5)$P_2$ following inhibition of different lipid kinases. Unexpectedly, this also revealed the presence of residual PI(3,5)$P_2$ positive membranes in the lumen of swollen lysosomes following PIKfyve inhibition using Apilimod. While the formation of these swollen lysosomes is well known (Cai et al., 2013), the nature of these intraluminal vesicles is less clear. Importantly, these intraluminal vesicles are inaccessible to cytosolic kinases and phosphatases. Thus, since PIKfyve is the only known kinase that generates PI(3,5)$P_2$, these pools must be present prior to Apilimod treatment and sequestered into these terminal lysosomes where they cannot be broken down. Further, PIKfyve forms a constitutive complex with the scaffolding protein Vac14 and the lipid phosphatase Fab1 that catalyzes 5′ dephosphorylation of PI(3,5)$P_2$ back to PI(3)P (Lees et al., 2020). This indicates an intricate crossregulation of the PIKfyve:Vac14:Fab1 complex to avoid futile phosphorylation cycles. In this context, it is tempting to speculate that the inhibition of PIKfyve might also indirectly influence the activity of Fab1. Further work into this regulation is underway, and the ability to stain for phosphoinositides and visualize intraluminal membranes will be crucial toward its progress.

Taken together, we believe that this toolkit is a valuable addition to the repertoire of cell biologists to investigate phosphoinositide biology. Finally, this approach has the potential to reveal novel insights related to health and disease, especially in cell types and models that cannot be transfected, such as patient-derived primary cells and tissue sections.

## Materials and methods
### Constructs and cloning
DNA for all biosensors was optimized for bacterial expression, synthesized, and cloned into standard ITPG-inducible bacterial expression vectors containing Kanamycin resistance. Following the N-Terminal 6xHis tag, an HRV3C cleavage site was added upstream of the SNAP tag for the optional removal of the 6xHis tag. All constructs (apart from the PI(5)P biosensor) are available through Addgene (https://Addgene.org/Hannes_Maib).

### Purification and labeling of recombinant biosensors
All recombinant biosensors were expressed in BL21 bacterial cells using standard approaches. In our hands, recombinant probes expressed strongly and 4L of bacteria were sufficient to generate ~1 ml of ~20–50 µM recombinant protein, generating enough material for several hundred stainings (at 100 µl per coverslip at 500 nM final concentration). Transfected bacteria were grown in LB containing 1.75 wt/vol % lactose and antibiotic

at 37°C to OD600 = 0.8, whereupon temperature was lowered to 18°C for 10–12 h. Alternatively, expression can be induced by the addition of 0.5 mM ITPG at OD600 of ~0.8 and grown at 18°C overnight. Cells were pelleted, resuspended in standard buffer (20 mM HEPES, 250 mM NaCl, and 0.5 mM TCEP), lysed using ultrasound sonication, and clarified by centrifugation at 55,000 × $g$ for 1 h at 4°C. Cleared lysates were passed through a 0.45-µM filter (Sartorius), and protein was purified by Ni$^{2+}$ affinity chromatography using 5 ml His-Trap HP column (Cytiva) against an increasing gradient of standard buffer containing 250 mM Imidazole. Peak fractions were pooled and resuspended in a buffer containing 20 mM HEPES and 0.5 mM TCEP, resulting in a final NaCl concentration of ~50 mM, and further purified by anion exchange on a 5-ml Capto-Q or Capto-S column (Cytiva) (depending on the pI of the biosensor) against a gradient of 20 mM HEPES and 0.5 mM TCEP containing 1 M NaCl. Peak fractions were pooled and purified by size-exclusion chromatography using a Superdex 200 16/60 pg column (Cytiva) in a standard buffer. All proteins were aliquoted, frozen in liquid nitrogen, and stored at –80°C. Aliquots were labeled with SNAP-Surface Alexa Fluor 488, 546, or 647 (NEB) or JF650 (Janelia Fluor) in a 2:1 M excess for 2–3 h on ice in a standard buffer. Excess dye was removed by dialysis against standard buffer at 4°C overnight. Note that the removal of excess dye improves the signal-to-noise of cellular staining but is not a strict necessity as it gets washed out during the staining process. For cleavage of the 6xHis tag, peak fractions were incubated with recombinant PreScission protease for 1 h at 4°C.

### In vitro lipid-binding experiments
1 mg liposomes, containing each of the eight different phosphoinositides, were produced by mixing 95 mol % 1-palmitoyl-2-oleoyl-sn-glycero-3-phosphocholine with 5 mol% of the respective phosphoinositides together with 0.1% Atto647N-DOPE. The mixtures were evaporated under nitrogen and dried overnight in a vacuum extruder. Dried lipids were resuspended in 1 ml buffer containing 150 mM NaCl, 20 mM HEPES, and 0.5 mM TCEP and subjected to six freeze–thaws in liquid nitrogen. Liposomes of ~100 nm diameter were generated by passing the lipid mixture 11 times through a 100-nm filter (Whatman Nuclepore). Liposomes were aliquoted, snap-frozen, and stored at –20°C.

Membrane-coated beads were generated by adding 10 µg of liposomes to ~0.5 × 10$^6$ 10-µm silica beads (Whitehouse Scientific) in 100 µl of 200 mM NaCl for 30 min rotation at room temperature. Beads were washed twice and resuspended in a buffer containing 150 mM NaCl, 20 mM HEPES, and 0.5 mM TCEP and blocked with 200 µg/ml β casein. Drops of 10 µl beads were added into the corners of uncoated µ-Slide eight-well chambers (Ibidi), and 10 µl of the purified biosensors was added to 200 nM final concentration. Lipid-binding kinetics were allowed to equilibrate for 30 min at room temperature before imaging close to the equator of the beads.

Confocal images were acquired using a Leica SP8 Confocal Microscope with a Leica HC PL APO CS2 63×/1.40 oil objective at 0.75 base zoom with 1,024 × 1,024 pixels scan. Alexa488 and Atto647N-DOPE fluorescence were imaged simultaneously

without any measurable bleed-through. Data were analyzed using a custom ImageJ script that segments the Atto647N-DOPE channel to create a mask around the outer circumference of each bead. Segmented masks were then used to measure Alexa488 fluorescence around each bead.

## Cell culture
HeLa cells and MIA-Paca2 cells were cultured in DMEM supplemented with 10% FCS and penicillin/streptomycin and passaged every 2–3 days. NMuMG cells were grown in DMEM with 4.5 g/liter glucose and 10 mcg/ml insulin and 10% FCS. For the generation of 3D spheroids, single-cell suspensions were seeded in eight-well glass bottom imaging chambers (Ibidi) on a cushion of Matrigel (BD Bioscience) in 2% Matrigel containing growth media for 5 days before being fixed and stained. HeLa cells were transfected using PolyFect transfection reagent for over-expression of genetically encoded biosensors and grown on ibidi µ-Dish 35 mm, high Grid-500 for relocalization. The HeLa cell line was a kind gift from Prof. Elizabeth Smythe (University of Sheffield, Sheffield, UK) and the Mia-Paca2 cell line was a kind gift from Prof. Jason King (University of Sheffield, Sheffield, UK). The NMUMG cell line was a kind gift from Dr Elena Rainero (University of Sheffield, Sheffield, UK).

## Cell staining using recombinant biosensors
Cells were grown in six-well dishes on 24 × 24 mm 1.5 glass coverslips to ~80% confluence in full growth media. Media was removed and 4% PFA with 0.2% Glutaraldehyde (both EM grade) in PBS prewarmed to 37°C was added for 20 min at room temperature followed by two quick washes and incubation with 50 mM $NH_4Cl$ in PBS for 20 min. Afterward, the six-well dishes with coverslips were transferred onto ice, and the cells were washed with ice-cold PIPES buffer. To make up the PIPES buffer, 3 g piperazine-$N,N'$bis[2-ethanesulfonic acid] was dissolved in 1 M NaOH, the volume was adjusted to 1 L with $ddH_2O$, 0.58 g NaCl and 0.2 g $MgCl_2$ hexahydrate were added, and the pH was adjusted to 7.2. Cells were stained, blocked, and permeabilized at the same time in 5% BSA + 0.5% (vol/wt) Saponin (S7900 from quillaja bark; Sigma-Aldrich) in PIPES with the addition of labeled biosensors to a final concentration of 500 nM for 45–60 min on ice. For staining, a standard metal heat block was inverted and cooled down in an ice bucket. A small piece of parafilm was added onto the inverted heat block, 100 µl drops of the staining solution was applied to the parafilm, and the coverslips were placed on top (with the cells facing the liquid). Afterward, the coverslips were transferred back into the six wells on ice and washed three times with ice-cold PIPES. Cells were post-fixed with 2% PFA in PBS on ice for 10 min before being returned to room temperature for another 10 min. Cells were washed with 50 mM $NH_4Cl$ in PBS at room temperature, rinsed with $ddH_2O$, mounted, and sealed using Prolong Gold.

## Transferrin pulse-chase assay
HeLa cells were grown on coverslips and chilled on ice for 10 min in ice-cold PBS. Transferrin conjugated to Alexa568 (Thermo Fisher Scientific) was bound to the cell surface at 5 µg/ml for 30 min in PBS on ice 5 µg/ml followed by three wash steps

on ice. Cells were transferred into prewarmed (37°C) full growth media for indicated timepoints before being fixed and stained against PI(3)P and PI(4)P using recombinant biosensors conjugated to either Alexa546 or Alexa488 as described above. Cells were imaged using LSM980 Airyscan with a Plan Apochromat 63 × 1.4NA and processed using the joint deconvolution 3D Airyscan processing. Data were analyzed by manually cropping out single cells and measuring the signal overlap between transferrin and PI(3)P or PI(4)P using the automated segmentation pipeline Squassh (Rizk et al., 2014).

## Staining of *Drosophila* pupal wings
Wings were dissected from w[1118] (FlyBase: FBal0018186) pupae and aged for 28 h after prepupa formation. These were fixed for 30 min in 4% paraformaldehyde/0.2% glutaraldehyde in PBS at room temperature, the outer cuticle was removed, and then quenched for 15 min in 50 mM $NH_4Cl$ in PBS.

Subsequent staining steps were carried out on ice or at 4°C. Wings were blocked and permeabilized for 45 min in 5% BSA/ 0.5% Saponin in PIPES buffer. Recombinant biosensors were diluted in 0.1% Saponin in PIPES buffer (PI(4)P biosensor at a final concentration of 150 nM and PI(4,5)$P_2$ biosensor at 500 nM) with 1:200 Alexa488-conjugated Phalloidin (Invitrogen) and incubated with the wings for 1 h. Wings were then washed 12 times in 0.1% Saponin in PIPES buffer, post-fixed for 10 min in 2% PFA, washed a further four times with 0.1% Saponin in PIPES buffer, and mounted in 2.5% DABCO with 10% glycerol in PBS. The wings were imaged on a Nikon A1R GaAsP confocal microscope using a 60× NA1.4 apochromatic lens within 1 week of mounting. Images were acquired posterior to longitudinal vein 4 with a pixel size of 110 nm and z-sections spaced by 150 nm.

## Microscopy
Single-color staining using the recombinant biosensors conjugated with Alexa488 (Fig. 3) was carried out using LSM880 Airyscan confocal microscope with a Plan Apochromat 63× 1.4 NA followed by standard 2D Airyscan processing. Multiplex staining using recombinant biosensors conjugated with Alexa488, 546, or 647 was carried out in full Z-Stacks of 150 nm step size using LSM980 Airyscan with a Plan Apochromat 63 × 1.4NA and processed using the joint deconvolution 3D Airyscan processing.

STED imaging was performed on a commercial expert line Abberior STED microscope equipped with 485, 561, and 645 nm excitation lasers and an Olympus Objective UPlanSApo 100× 1.40 NA. For two-color STED of Arf1-Halo and PI(4)P, cells were seeded on a glass-bottom dish (3.5 cm diameter, no. 1.5 glass; Cellvis) coated with fibronectin (Sigma-Aldrich). Labeling with Halo substrate JF571 was carried out for 1 h using 1 µM stocks. After the staining, cells were washed three times with growth medium to get rid of the excess dye and left for 1 h in an incubator at 37°C and 5% $CO_2$. Cells were then fixed and stained with biosensors against PI(4)P as described above. The 775-nm depletion laser was used to deplete dyes in two-color STED experiments. Two-color images were recorded sequentially line by line. Detection was carried out with avalanche photodiodes with detection windows set to 571–630 nm and 650–756 nm. The acquisition was carried out with Instruments Development

Team, Imspector Image Acquisition & Analysis Software v16.3.16118 (https://www.imspector.de). The pixel size was set to 20 nm, and Raw STED images were deconvolved to reduce noise using Richardson–Lucy deconvolution from the Python microscopy PYME package (https://python-microscopy.org/).

### Electron microscopy
MIA-Paca2 cells were grown in media containing 10% FCS, treated with 1 μM Apilimods in DMSO for 1 h, fixed in 4%PFA with 0.2% glutaraldehyde, pelleted, and embedded in a resin according to standard protocols. Pellets were stained with 1% osmium for 1 h and 0.5% uranyl acetate overnight, embedded in epon resin, cut into 70-nm sections, and stained en bloc with uranyl acetate and lead citrate before imaging using an FEI Tecnai T12 Spirit at 80 kV.

### Online supplemental material
Fig. S1 shows the staining using recombinant biosensors of cells ectopically overexpressing biosensors. Fig. S2 shows the staining using recombinant biosensors following the inhibition of lipid kinases. Fig. S3 shows the staining processes using the biosensor against PI(4,5)P$_2$ following the removal of the His tag as well as a one-step purification protocol.

### Data availability
The data are available from the primary corresponding author (Hannes Maib h.maib@sheffield.ac.uk) upon request.

## Acknowledgments
We acknowledge the Wolfson Light Microscopy Facility for the use of the LSM 980 Airyscan 2 supported by the Medical Research Council grant MR/X012077/1 and Christopher J. Hill from the University of Sheffield Faculty of Science EM facility for support with this section electron microscopy. We thank Anja Heuhsen for help with sample preparation for STED.

H. Maib is supported by a Wellcome Trust early Career award 225528/Z/22/Z. We further thank Prof. Jason King and Prof Elizabeth Smythe for their helpful comments on the manuscript. R. Hunton is supported by BBSRC DTP studentship (BB/M011151/1) and DS is supported by a Wellcome Trust Senior Fellowship (210630/Z/18/Z). D.H. Murray is supported by the Wellcome Trust grant 211193/Z/18/Z. P. Adarska is supported by the Forschungsgemeinschaft (German Research Foundation) grant SFB/TRR186 (Project A20). Open Access funding provided by University of Sheffield.

Author contributions: H. Maib conceived the project and designed and performed the experiments. P. Adarska and F. Bottanelli performed and supervised the STED imaging. R. Hunton and D. Strutt performed and supervised the imaging of *Drosophila* pupal wing. J. Vines performed the live imaging of PI(3,5)P$_2$. D.H. Murray supervised the cloning, purification, and in vitro verification of recombinant biosensors. H. Maib and D.H. Murray prepared the manuscript, which was edited and approved by all other authors.

Disclosures: The authors declare no competing interests exist.

Submitted: 23 October 2023

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

# Supplemental material

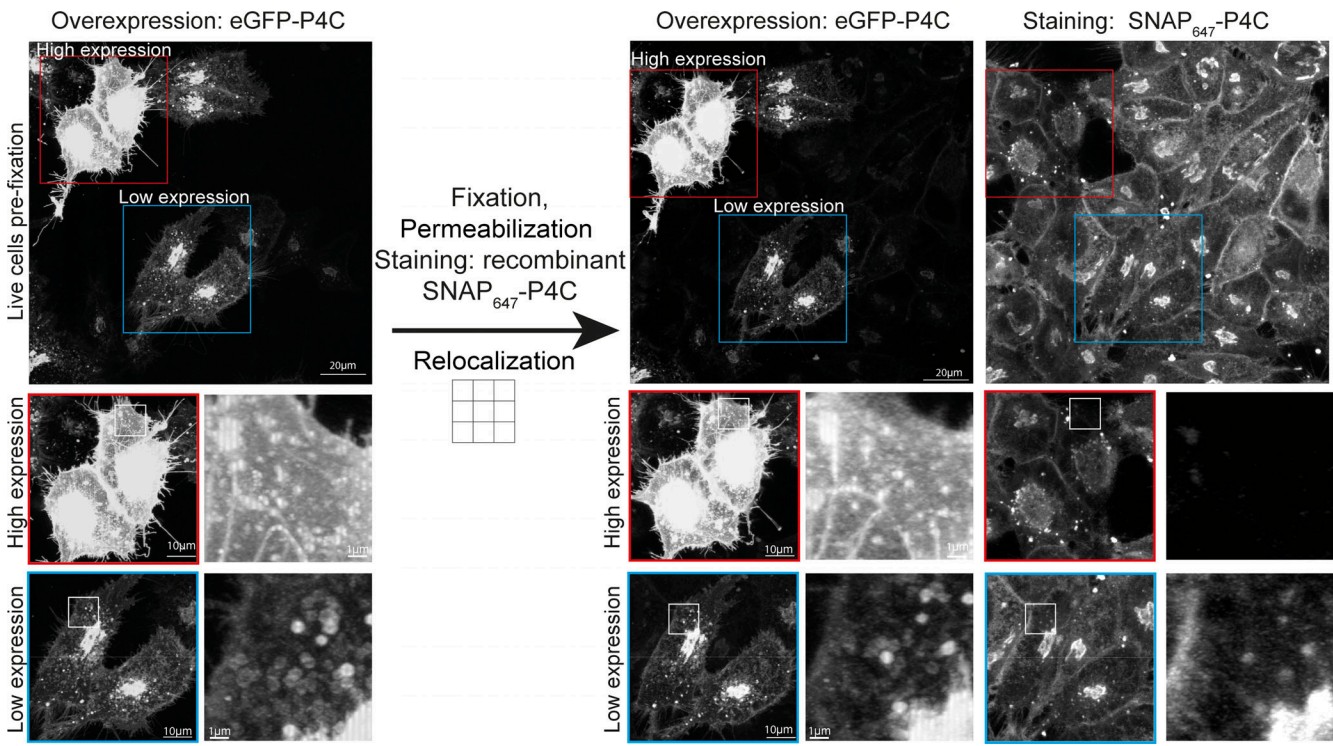

Figure S1. **Staining of cells ectopically overexpressing biosensors.** Hela cells were transfected with plasmids for ectopic overexpression of eGFP-tagged P4C to detect PI(4)P. Cells were seeded on a gridded glass coverslip and imaged live before being fixed, permeabilized, and stained against PI(4)P using a recombinant biosensor labeled with Alexa647. The same cells were relocalized and imaged again. Inserts highlight cells with high and low expression levels of the PI(4)P biosensor. Images are maximum intensity projection of whole z-stacks.

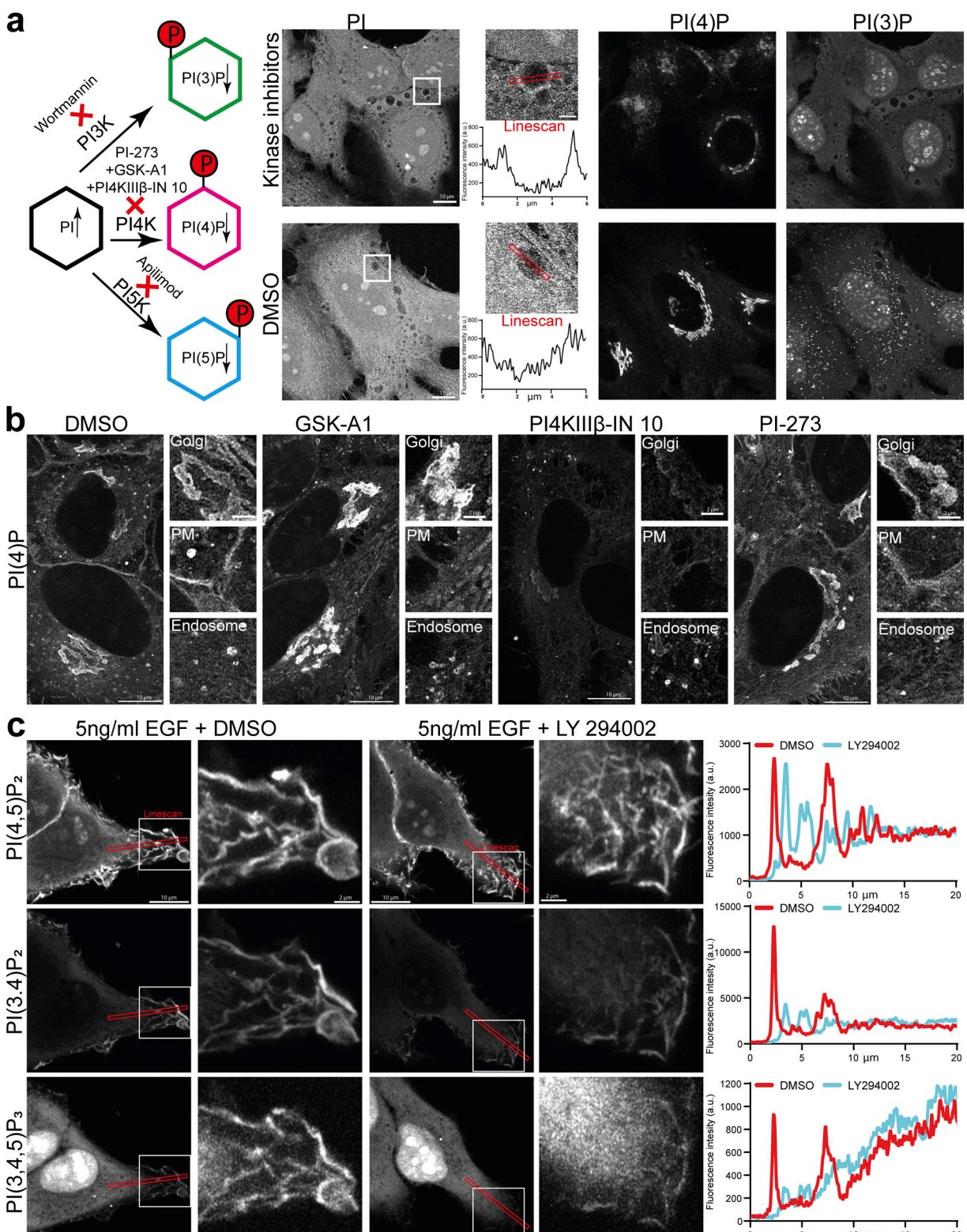

Figure S2.   **Phosphoinositide staining following inhibition of lipid kinases. (a)** HeLa cells were treated with a kinase inhibitor cocktail of Wortmannin, PI-273, GSK-A1, PI4KIIIβ-IN 10, and Apilimod, each at 500 nM for 30 min before being fixed and stained using recombinant biosensors against PI, PI(4)P and PI(3)P conjugated to Alexa488, 546, or 647, respectively. Linescan shows the signal of PI on the membrane of large vacuoles that can be found throughout the cell. **(b)** HeLa cells were treated with either GSK-A1, PI-273, or PI4KIIIβ-IN 10 at 100 nM each for 30 min before being fixed and stained against PI(4)P using recombinant biosensors against PI(4)P conjugated to Alexa488. **(c)** HeLa cells were treated with LY 294022 at 1 μM for 30 min, stimulated with 5 ng/ml human EGF for 10 min, and fixed and stained using recombinant biosensors against PI(4,5)P$_2$, PI(3,4)P$_2$, and PI(3,4,5)P$_3$ conjugated to Alexa488, 546, or 647, respectively. Linescans show signals of PI(4,5)P$_2$, PI(3,4)P$_2$, and PI(3,4,5)P$_3$ in DMSO and LY294002 treated cells at membrane ruffles as well as the cytosolic background.

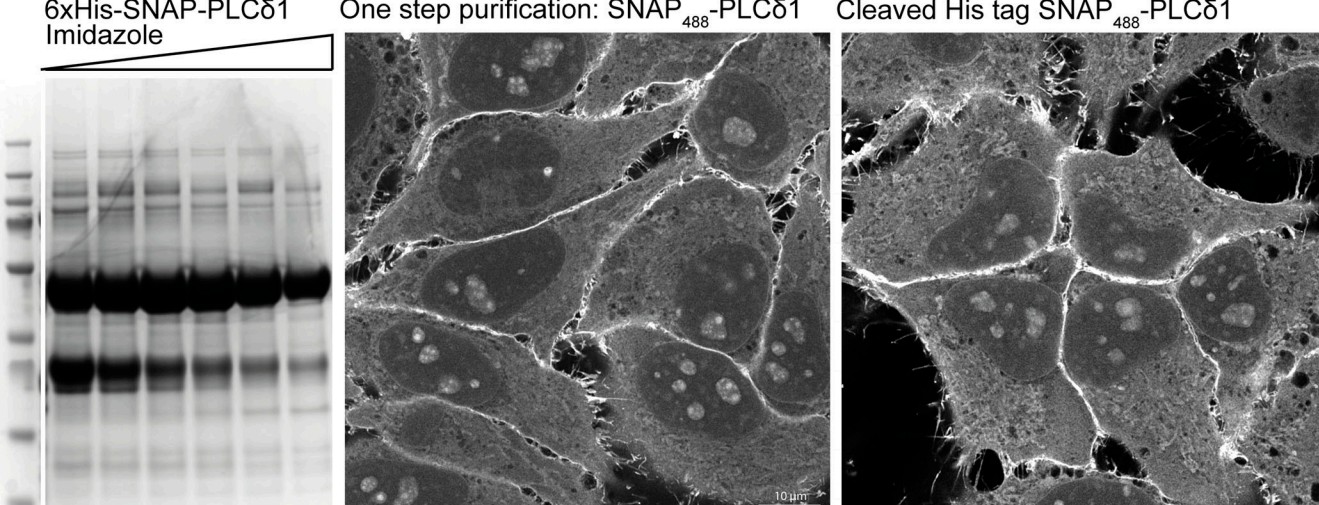

Figure S3. **Staining using one step His purification.** The recombinant biosensor against PI(4,5)P$_2$ (6xHis-SNAP-PLCδ1) was purified from BL21 E. coli using a HisTrap column and eluted against an increasing gradient of 250 mM Imidazole. Peak fractions were pooled and either directly labeled with SNAP-Surface Alexa488 or first incubated with recombinant PreScission Protease to cleave the 6xHis tag and used to stain HeLa cells as previously described. Source data are available for this figure: SourceData FS3.

