## [Peer Review File · The Journal of Cell Biology]

Recombinant biosensors for multiplex and super-resolution imaging of phosphoinositides

Hannes Maib, Petia Adarska, Robert Hunton, James Vines, David Strutt, Francesca Bottanelli, and David Murray

Corresponding Author(s): Hannes Maib, University of Sheffield

Review Timeline:	Submission Date:	2023-10-23
	Editorial Decision:	2023-11-21
	Revision Received:	2024-02-16
	Editorial Decision:	2024-03-06
	Revision Received:	2024-03-08

Monitoring Editor: Tamas Balla

Scientific Editor: Andrea Marat

Transaction Report:

DOI: <https://doi.org/10.1083/jcb.202310095>

November 21, 2023

Re: JCB manuscript #202310095

Dr. Hannes Maib
University of Sheffield
Biosciences
Western Bank
Sheffield S10 2TN
United Kingdom

Dear Dr. Maib,

Thank you for submitting your manuscript entitled "Recombinant biosensors for multiplex and super-resolution imaging of phosphoinositides". The manuscript was assessed by expert reviewers, whose comments are appended to this letter. We invite you to submit a revision if you can address the reviewers' key concerns, as outlined here.

You will see that the reviewers all appreciate the potential importance of the phosphatidylinositol probes you have developed. However, they have suggested several controls to further characterize and validate these probes. We agree that they represent a valuable addition to the currently available techniques to detect phosphatidylinositols, and therefore a suitably revised study will make an impactful JCB Tool. To ensure they are published for the community in a timely manner, we find the following essential to address:

- As noted by the reviewers, your probes only reliably detect 6 and not all 8 phosphatidylinositols, which should be clarified. Please keep PI5P in the manuscript as a demonstration it does not work. The PI probe should be more thoroughly tested as indicated by reviewer 2.
 - The specificity of stainings and their further characterization using commercially available inhibitors suggested by reviewer 2 must be attempted.
 - To ensure the probes can be easily used by others clarify all aspects of the methodology in the materials and methods. You can also consider publishing a step-by-step protocol in an appropriate venue. Looking into commercialization is an interesting suggestion but is not relevant at the current stage of revision.
 - Test for potential impacts of the His tag.
 - Ensure limitations of the probes are clearly discussed for non-experts, for example that multiplexing with various fluorophores should not be used to compare relative quantities.

 - Regarding the suggestion of reviewer 1 to compare your recombinant probes with conventional probes, each method of course has potential caveats and limitations. Therefore, we do not find comprehensive side-by-side comparisons necessary, and we would not want such comparisons to give a potentially false impression of one method being inherently superior as this may vary depending on the application. As a further validation of your probes we suggest instead to examine how expression of a biosensor (e.g. PLC δ 1PH or Tubby tagged with a different fluorophore than the one used on the recombinant probe) would affect the staining with the probe using the protocol. For example, in cells expressing the biosensor is the lipid pool upregulated resulting in a larger signal with the fixed staining. This should also indicate if a protein that occupies a lipid occludes it from detection, as well as providing a comparison of distribution of expressed versus stained fixed cells using the recombinant probe.
- Finally, please respond to all other points via text edits and additional discussions where appropriate. Specifically:
- While we would welcome data to provide a further *in vitro* characterization of the probes as well as a comparison of the various fluorophore coupling and brightness efficiency as suggested by reviewer 3, this is not essential.
 - We also believe that comparing your method with antibodies would not be essential at this time as few inositol lipid antibodies have been thoroughly vetted for their specificity and accuracy.

GENERAL GUIDELINES:

Text limits: Character count for an Tools is < 40,000, not including spaces. Count includes title page, abstract, introduction, results, discussion, and acknowledgments. Count does not include materials and methods, figure legends, references, tables, or supplemental legends.

Figures: Tools may have up to 10 main text figures. Figures must be prepared according to the policies outlined in our

Instructions to Authors, under Data Presentation, <https://jcb.rupress.org/site/misc/ifora.xhtml>. All figures in accepted manuscripts will be screened prior to publication.

IMPORTANT: It is JCB policy that if requested, original data images must be made available. Failure to provide original images upon request will result in unavoidable delays in publication. Please ensure that you have access to all original microscopy and blot data images before submitting your revision.

Supplemental information: There are strict limits on the allowable amount of supplemental data. Tools may have up to 5 supplemental figures. Up to 10 supplemental videos or flash animations are allowed. A summary of all supplemental material should appear at the end of the Materials and methods section.

Please note that JCB now requires authors to submit Source Data used to generate figures containing gels and Western blots with all revised manuscripts. This Source Data consists of fully uncropped and unprocessed images for each gel/blot displayed in the main and supplemental figures. Please be sure to provide one Source Data file for each figure that contains gels and/or blots along with your revised manuscript files. File names for Source Data figures should be alphanumeric without any spaces or special characters (i.e., SourceDataF#, where F# refers to the associated main figure number or SourceDataFS# for those associated with Supplementary figures). The lanes of the gels/blots should be labeled as they are in the associated figure, the place where cropping was applied should be marked (with a box), and molecular weight/size standards should be labeled wherever possible.

The typical timeframe for revisions is three to four months. While most universities and institutes have reopened labs and allowed researchers to begin working at nearly pre-pandemic levels, we at JCB realize that the lingering effects of the COVID-19 pandemic may still be impacting some aspects of your work, including the acquisition of equipment and reagents. Therefore, if you anticipate any difficulties in meeting this aforementioned revision time limit, please contact us and we can work with you to find an appropriate time frame for resubmission. Please note that papers are generally considered through only one revision cycle, so any revised manuscript will likely be either accepted or rejected.

Thank you for this interesting contribution to Journal of Cell Biology. You can contact us at the journal office with any questions at cellbio@rockefeller.edu.

Sincerely,

Tamas Balla, MD, PhD
Monitoring Editor

Andrea L. Marat, PhD
Senior Scientific Editor

Journal of Cell Biology

Reviewer #1 (Comments to the Authors (Required)):

Phosphoinositides are lipids involved in a variety of cellular functions, especially signaling and membrane trafficking. While genetically-encoded biosensors and antibodies for different phosphoinositides have been reported, they all come with their advantages and limitations, for example artifacts generated by overexpression of genetically encoded sensors.

In this manuscript, Maib and coworkers report a series of recombinant biosensors to visualize cellular phosphoinositide pools which should overcome some of the limitations of other biosensors. To this end, they produce a series of well-described phosphoinositide-binding proteins/ known biosensors fused to the SNAP-tag self-labelling system as recombinant proteins. Importantly, each of the biosensors is tested in vitro for its specificity, and the functionality is demonstrated in cells, spheroids and Drosophila wings. They further show the suitability of their system for superresolution microscopy, and exploit the

availability of multiple dyes for the SNAP-Tag for easy multiplex imaging.

This series of biosensors offers an interesting new tool to visualize phosphoinositide pools. The authors thoroughly characterize their specificity, and the detailed purification and labelling protocols make it an easily accessible resource for cell biologists. However, there are several concerns which need to be addressed. Specifically, the manuscript misses critical controls and benchmarking comparisons of the new toolbox to existing sensors.

a) This reviewer questions the wisdom of including the ING2-based PI5P probe. The authors themselves show that this is not a specific probe, and to include it will likely lead to reports of purely artefactual PI5P pools by less discerning users. The literature is already filled with articles based on low specificity antibodies, and it feels wrong to add another blunt tool to this.

b) It is not clear if the recombinant probes can indeed report the majority of biologically significant phosphoinositide pools. In a living cell, phosphoinositide binding proteins would likely occupy a large part of the available phosphoinositide pools and compete with each other for binding. As the authors clearly describe, this is one of the downsides of expressed biosensors, as they can outcompete the endogenous effectors. However, it is not clear to what extent any free pools exist which can be detected by the recombinant sensors in fixed cells. The majority of available phosphoinositides would likely be occupied by their effectors, and the fixation procedure employed here (PFA with 0.2% GA) would strongly cross-link any endogenous phosphoinositide-binding protein and presumably sequester the associated phosphoinositide pool. Thus, it is critical to test the localization of their recombinant probes in cells (weakly) expressing "conventional" phosphoinositide probes to see if these report the same pools.

c) Along the same lines as the previous point, the permeabilization procedure needs to be controlled. Treatment with a permeabilizing agent - even a quite mild one like Saponin, can strongly alter the membrane. The authors themselves report that permeabilization can alter the behavior of a PI4P-binding antibody, and then report that their probe faithfully reports several pools reported by the antibody. However, there is no data comparing their probes to this antibody using their fixation procedure. To verify that their permeabilization procedure does not affect phosphoinositide composition, it is critical to compare their method to a non-detergent based permeabilization, e.g. mechanical unroofing of the cells, or to cells expressing biosensors before and after fixation.

Both this point and the point above are critical benchmarks to compare the new set of biosensors to existing tools, so that potential users can decide on the best tool for their purpose.

Reviewer #2 (Comments to the Authors (Required)):

This manuscript presents a new suite of tools for staining phosphoinositides in fixed cells. The manuscript details 8 probes with the simple and versatile SNAP-tagging technology, showing the desired selectivity for specific PI lipids on artificial membranes for 7 of them. It then goes on to show staining for all in HeLa cells with a unified staining protocol that simplifies a previously published method. Super-resolution imaging of PI3P and PI4P probes is then shown using STED. Multiplexed imaging is then shown for two or three probes in HeLa cells, 3D NMuMG cultures and Drosophila pupal wings, demonstrating applicability to complex systems such as tissue sections and organoids, currently not accessible with transfected biosensors. Next, the authors show multiplexed imaging of PI4P, PI3P and PI(3,5)P2, showing that PI(3,5)P2 is present in intraluminal vesicles that enlarge after PIKfyve inhibition. Collectively, these data show the utility of these probes and demonstrate their capacity to provide insights not possible with transfected probes. As such, I think this satisfies the core criteria for a methods paper and as such, I think this paper is suitable for publication as a Tools paper in JCB.

That said, whereas the data by and large look convincing and correct, the paper would be strengthened by some additional characterization that would underscore the selectivity of staining in fixed cells. Specific suggestions are provided below. All would be simple experiments to be conducted using the current assays with existing or commercially available reagents, and could be accomplished within a reasonable revision period.

Conceptual comments

The abstract states "Here, we present a toolkit for the reliable, fast, multiplex, and super-resolution detection of all 8 phosphoinositides". However, I found myself slightly disappointed when reading the manuscript, because that convincing labeling is only demonstrated for 6 of these. The PI5P probe is not selective on the membrane-coated beads, and the PI probe does not produce convincing labeling (more on that below). Therefore, the abstract should be amended, perhaps to say something more diplomatic, like "detection of all phosphoinositide species for which suitable probes exist". I was personally bummed there was not a good probe for PI5P. This is not the authors fault, but I had a niggling hope given their recent discovery of the long elusive PI(3,5)P2 probe!

The PI probe: the diffuse staining pattern is inconclusive. This could be caused by the ubiquity of the lipid in intracellular membranes as the authors suggest, but it could also be non-selective. I have two suggestions that could shed light on which it is: The most trivial is that Pemberton et al (and Zewe et al) reported ubiquitous, widespread PI distribution - but not in the plasma membrane. So, this might be conveniently shown by multiplexed imaging with the PH-PLCD1 probe. Secondly, some demonstration that the labelling is PI-dependent might help. One idea is to use a bacterial PI-PLC protein (the Bacillus protein

can be bought from Sigma) in the fixed cells prior to staining to deplete the PI, maybe during the blocking step. Ice-cold temperatures may inhibit the enzyme, though I have had good experience with other bacterial enzymes working on ice so I think it is worth a try.

The PI3P probe: The staining looks very convincing, but to support the statement of "clear labelling of endo/lysosomal structures throughout the cell", co-localization with markers should be used. This may even be doable after the post-fixation step to stain for markers such as EEA1 or LAMP1. Alternatively, the authors could use expressed markers such as Rabs, or even endogenously tagged proteins where available (as they did for Arf1). Perhaps the simplest approach would be to show co-localization with the expressed EGFP-FYVE-HrsX2!

The PI4P probe: As for PI3P, some co-localization should be provided to substantiate the claim of localization to "trafficking vesicles". Late endocytic and recycling compartments are the two that have been reported in the literature. The Golgi labelling is substantiated in Figure 4. The plasma membrane staining is also somewhat hard to make out (compared to GFP-SidC when expressed). Are there clearer examples?

In addition, selectivity of the pools can be demonstrated by elimination of specific pools using selective PI4K inhibitors, which are commercially available. These include PM PI4KIIIalpha (GSK-A1, Sigma), PI-273 for endosomal PI4KIIalpha and PI4KIIIbeta-IN-10 for Golgi PI4KIIIbeta (both MedChemExpress).

PIP3 and PI(3,4)P2 probes: The majority of these lipids are synthesized by class I PI3K, for which there are many commercially available, potent inhibitors (e.g. GDC-0941). Depletion of the staining by these compounds should be shown as a minimum standard for specificity in cells. Furthermore, given that these lipids are predominantly formed during receptor signaling events, enhanced production following cellular stimulation with e.g. EGF or insulin in HeLa cells would enhance the confidence in the probes.

All probes: Inclusion of millimolar concentrations of neomycin in the blocking and staining solutions will most likely block all the polyphosphoinositides, and be another nice specificity control. I wouldn't be surprised if it also worked for the PI probe too! This would be a very easy control experiment.

The manuscript states in the results that: "PI(3,5)P2 on contrast is located on the same membrane as PI(3)P". As I read this, I was struck by the notion that every cartoon in every review I have read on this topic has PI3P and PI(3,5)P2 defining distinct membrane compartments in the cell (normally, early vs late endocytic compartments). Given the substrate for PIKfyve, of course they would co-localize. But as the first demonstration of this fact, I think the authors could emphasize this point more, perhaps in the discussion on the ramifications for defining late vs early endosomal identity with these lipids.

Value as a tool/method: This protocol can indeed be easily implemented by other labs, justifying its publication in a high profile journal. Distribution of the constructs in Addgene is a requisite for this, and I commend the authors for doing this. An additional suggestion (though not a pre-requisite for publication) is to see if a company is interested in producing the proteins for distribution commercially? Echelon Biosciences do sell some GST-tagged probes and may be interested in providing this as a service with the probes reported (assuming there are no licensing issues with the use of SNAP, for example). Commercial availability would, I suspect, greatly enhance adoption of the approach.

Technical comments

Figure 2: the number of beads quantified is stated, but this should be supported by the number of independent experiments that these were quantified from.

The source and validity of the cell lines used should be indicated in the methods.

The composition of the PIPES buffer is not stated. This could be important!

What is the source of the saponin used? This is a reagent that can be quite variable given the source, so stating the manufacturer and catalogue number will maximize reproducibility.

Reviewer #3 (Comments to the Authors (Required)):

The authors of this study generated plasmids that enable expression, purification and fluorescent labeling of biosensors for phosphoinositides. Fluorescent protein tagged biosensors have been reported and the use of a SNAP tag increases flexibility in the use of fluorescent labels. The biosensors were characterised *in vitro* and subsequently used to image phosphoinositides at high resolution in fixed cells, spheroids and drosophila pupal wings.

My main worry is that it is unclear how the lipids are fixed by crosslinking with formaldehyde. Next to this (although the selectivity is nicely tested in an *in vitro* system) it is not clear to what extent the biosensors exhibit aspecific binding to cellular components.

Some evidence of aspecific binding seems to present in the images in figure 3, i.e. labeling of nucleoli for several of the probes and the reticular labeling observed with the PtdIns(3,4,5)P3 probe seems aspecific as well. Below I will list my comments that hopefully can be used to improve this work.

1. In the abstract "we present a toolkit for the reliable, fast, multiplex, and super-resolution detection of all 8 phosphoinositides" the use of "multiplex" here suggests (at least, this is what I understood before reading the entire paper) that it is possible to image all 8 at the same time. I recommend to rephrase the sentence to avoid this suggestion.

2. The authors state that "liposomes composed of 95% 1-palmitoyl-2-oleoyl-glycero-3-phosphocholine (POPC), and 5% of each phosphoinositide. These membrane coated beads are an ideal substrate to evaluate binding specificities, as they faithfully mimic the charge and fluidity of biological membranes (Pucadyil and Schmid, 2010)".

This statement raises several questions:

-The statement suggests that the coated beads mimic fluidity. In the study that is cited, DOPC/DOPS and DOPE were used, which have different acyl chains, and therefore a different fluidity. So, I'm not sure why this reference is used here and there might be better citations to support the idea that these beads "mimic the charge and fluidity of biological membranes".

-I would like to see citations to support the notion that 5% of phosphoinositides is a decent percentage. It's my understanding that PtdIns levels are much higher than PtdIns(3,4,5)P3. What at natural levels and how to these compare to the 5% used here?

-Since the charge increases with the number of phosphates on the inositol ring, there's a difference in charge between the PIs, most prominently between PtdIns(3,4,5)P3 and PI. Do they authors account for this in the in vitro assay? Would it make sense to include a control with POPC + POPA or POPS to study the effect of charge on binding of the biosensor?

3. From what I read, the His tag is still present. This may not be a problem, but did the authors study the effect of the His tag? Does the His tag affect specificity and may it explain some of the binding in the nucleus, e.g. for the PtdIns(4,5)P2 sensor?

4. The in vitro reconstitution and binding assay is nice, as it allows to directly observe the binding. In the data of figure 2 it is not clear whether these are from a single experiment, or from multiple replicates? In case of the second scenario, it would be more transparent to show the data as superplot.

5. Does the intensity presented in figure 2 reflect binding efficiency (I think so), and would it in that case be sensible to define a threshold for background binding? For instance, it seems like a value of 500 and anything below is background and the authors may consider showing this threshold as a line in the plots.

6. Related to the previous point, did the authors think of a control to determine background or aspecific binding in this assay?

7. In figure 3, DAPI seems to be included, but I do not see that mentioned in the legend or material and methods?

8. Several probes show localization in the nucleoli (if I interpret the images correctly). This should be clearly highlighted, because in the current manuscript it is only mentioned for PtdIns(5)P.

9. In figure 5C, the biosensors are used in Drosophila pupal wings. The PtdIns(4)P probe shows diffuse staining and it discussed that "Noticeably, PI(4)P is also enriched at these junctions albeit with a more diffuse staining pattern, indicating the presence of intracellular PI(4)P membranes in close proximity to the plasma membrane (Fig. 5c). These observations are in good agreement with the dynamic regulation of these two lipids at cell junctions in Drosophila tissue to drive recruitment of polarity proteins through electrostatic interactions (Dong et al., 2015; Lu et al., 2022)"

When I look at the paper by Lu et al (<https://elifesciences.org/articles/79582>), the localization of PtdIns(4)P with the SidM probe in drosophila is at the plasma membrane, with very little diffuse, intracellular labeling. This seems to be quite different from the diffuse staining that the author report here (figure 5C). Is the diffuse staining real, i.e. is it specific labeling of PtdIns(4)P? This needs to be discussed and compared to data from live cell imaging.

10. The authors generate biosensors with different fluorophores (Alexa488, 546 and 647) for multiplex imaging. Is anything known about the labeling efficiency and differences between the probes in terms of efficiency? In addition to using this for labeling on fixed cells, these probes could be used to strengthen the in vitro characterisation. For instance, a competition experiment between to similar binding domains with different labels. Or co-labeling of beads with two biosensors for different phosphoinositides, to show that two different lipids can indeed be detected on the same bead.

11. Related to the previous point, if a red biosensor (or unlabeled) effectively competes with a green biosensor on the beads (and that's expected), I would think that a similar strategy can be used in cells to verify specificity of the signal. For instance green and red biosensor can be mixed in a 1:9 ratio and in this situation the green channel would predominantly show the aspecific labeling.

12. In the end, this would be a toolkit that should be of use to others. This requires more detail in the materials and methods. For instance, what was the method for the e.coli lysis, what was the buffer for anion exchange and the size exclusion column, what

is the composition and pH of "PIPES buffer"? It would be informative to state the volume of e.coli culture that was used for the isolation and what the typical yields were obtained?

How much protein was labeled in what volume and for how many experiments can that be used (approximately).

13. Did the authors check whether a one step isolation with His tag is sufficient, as this would greatly improve the user-friendliness of the method?

Reviewed by Joachim Goedhart (University of Amsterdam).

Reviewer #1 (Comments to the Authors (Required)):

Phosphoinositides are lipids involved in a variety of cellular functions, especially signaling and membrane trafficking. While genetically-encoded biosensors and antibodies for different phosphoinositides have been reported, they all come with their advantages and limitations, for example artifacts generated by overexpression of genetically encoded sensors.

In this manuscript, Maib and coworkers report a series of recombinant biosensors to visualize cellular phosphoinositide pools which should overcome some of the limitations of other biosensors. To this end, they produce a series of well-described phosphoinositide-binding proteins/ known biosensors fused to the SNAP-tag self-labelling system as recombinant proteins. Importantly, each of the biosensors is tested in vitro for its specificity, and the functionality is demonstrated in cells, spheroids and Drosophila wings. They further show the suitability of their system for superresolution microscopy, and exploit the availability of multiple dyes for the SNAP-Tag for easy multiplex imaging.

This series of biosensors offers an interesting new tool to visualize phosphoinositide pools. The authors thoroughly characterize their specificity, and the detailed purification and labelling protocols make it an easily accessible resource for cell biologists.

However, there are several concerns which need to be addressed. Specifically, the manuscript misses critical controls and benchmarking comparisons of the new toolbox to existing sensors.

We thank the reviewer for the positive feedback on our approach and hope that we have addressed their concerns in the revised manuscript. Our key aim is to provide the community with an additional tool for the visualisation of phosphoinositides that should complement the existing methods. We find that each method has potential caveats and limitations and we do not want to give the false impression of one method being inherently superior, as this may vary depending on the application.

a) This reviewer questions the wisdom of including the ING2-based PI5P probe. The authors themselves show that this is not a specific probe, and to include it will likely lead to reports of purely artefactual PI5P pools by less discerning users. The literature is already filled with articles based on low specificity antibodies, and it feels wrong to add another blunt tool to this.

The reviewer is very right to point out the concerns of letting yet another blunt tool out into the community. However, we believe that it is due diligence to test the known biosensors against all eight phosphoinositides and also show the ones that do not work. In the revised version we have more clearly stated that the recombinant PI(5)P biosensor should not be used for this staining approach. To further discourage people from doing so, we have excluded this plasmid from our Addgene deposition.

b) It is not clear if the recombinant probes can indeed report the majority of biologically significant phosphoinositide pools. In a living cell, phosphoinositide binding proteins would likely occupy a large part of the available phosphoinositide pools and compete with each other for binding. As the authors clearly describe, this is one of the downsides of expressed biosensors, as they can outcompete the endogenous effectors. However, it is not clear to what extent any free pools exist which can be detected by the recombinant sensors in fixed cells. The majority of available phosphoinositides would likely be occupied by their effectors, and the fixation procedure employed here (PFA with 0.2% GA) would strongly cross-link any endogenous phosphoinositide-binding protein and presumably sequester the associated phosphoinositide pool.

Thus, it is critical to test the localization of their recombinant probes in cells (weakly) expressing "conventional" phosphoinositide probes to see if these report the same pools.

We thank the reviewer for this feedback and have done exactly as suggested. We have stained cells expressing the PI(4)P biosensor (at high and low levels) with the recombinant biosensor against the same phosphoinositide. As the reviewer predicted, high expression occupies most available phosphoinositide pools, occluding their detection. However low expression still allows for detection of PI(4)P on diverse subcellular membranes including the PM, Golgi and endosomes. This proves that the recombinant biosensors detects the same lipid pools as the genetically expressed probes. Thus, our approach is sensitive to the free population of phosphoinositides that are present by the time of fixation. This data has been added as Supplemental Figure S1. Also note that some of the faint vesicles that are weakly detected using expressed biosensors are difficult to detect using the recombinant probes. Thus (as with genetically expressed biosensors), minor pools might always escape detection due to being highly occupied by endogenous effectors. We have pointed out these limitations in the discussion section.

Figure S1: Staining of cells ectopically overexpressing biosensors. HeLa cells were transfected with plasmids for ectopic overexpression of eGFP tagged P4C to detect PI(4)P. Cells were seeded on a gridded glass cover slip and imaged live, before being fixed, permeabilised and stained against PI(4)P using recombinant biosensor labelled with Alexa 647. The same cells were relocalised and imaged again. Inserts highlight cells with high and low expression levels of the PI(4)P biosensor. Images are maximum intensity projection of whole Z-Stacks

c) Along the same lines as the previous point, the permeabilization procedure needs to be controlled. Treatment with a permeabilizing agent - even a quite mild one like Saponin, can strongly alter the membrane. The authors themselves report that permeabilization can alter the behavior of a PI4P-binding antibody, and then report that their probe faithfully reports several pools reported by the antibody. However, there is no data comparing their probes to this antibody using their fixation procedure.

To verify that their permeabilization procedure does not affect phosphoinositide composition, it is critical to compare their method to a non-detergent based permeabilization, e.g. mechanical unroofing of the cells, or to cells expressing biosensors before and after fixation. Both this point and the point above are critical benchmarks to compare the new set of biosensors to existing tools, so that potential users can decide on the best tool for their purpose.

Again, this is a very valid point and we hope to have addressed this concern in Supplemental Figure S1. We have expressed the biosensor against PI(4)P fused to eGFP in cells seeded on a grided glass coverslip and have imaged the very same cells prior to and after our staining approach. As can be seen by comparing the localisation of the PI(4)P biosensor in live and fixed cells, membrane organisation is unaltered following our staining approach. Further, fixation and permeabilisation still enables the detection of PI(4)P at the PM, Golgi and Endosomes.

Reviewer #2 (Comments to the Authors (Required)):

This manuscript presents a new suite of tools for staining phosphoinositides in fixed cells. The manuscript details 8 probes with the simple and versatile SNAP-tagging technology, showing the desired selectivity for specific PI lipids on artificial membranes for 7 of them. It then goes on to show staining for all in HeLa cells with a unified staining protocol that simplifies a previously published method. Super-resolution imaging of PI3P and PI4P probes is then shown using STED. Multiplexed imaging is then shown for two or three probes in HeLa cells, 3D NMuMG cultures and Drosophila pupal wings, demonstrating applicability to complex systems such as tissue sections and organoids, currently not accessible with transfected biosensors. Next, the authors show multiplexed imaging of PI4P, PI3P and PI(3,5)P2, showing that PI(3,5)P2 is present on intraluminal vesicles that enlarge after PIKfyve inhibition. Collectively, these data show the utility of these probes and demonstrate their capacity to provide insights not possible with transfected probes. As such, I think this satisfies the core criteria for a methods paper and as such, I think this paper is suitable for publication as a Tools paper in JCB.

That said, whereas the data by and large look convincing and correct, the paper would be strengthened by some additional characterization that would underscore the selectivity of staining in fixed cells. Specific suggestions are provided below. All would be simple experiments to be conducted using the current assays with existing or commercially available reagents, and could be accomplished within a reasonable revision period.

We thank the reviewer for the positive assessment of our manuscript and have included several new experiments to address the selectivity of our staining approach. We think this has considerably improved the manuscript and are genuinely grateful for the input.

Conceptual comments

The abstract states "Here, we present a toolkit for the reliable, fast, multiplex, and super-resolution detection of all 8 phosphoinositides". However, I found myself slightly disappointed when reading the manuscript, because that convincing labeling is only demonstrated for 6 of these. The PI5P probe is not selective on the membrane-coated beads, and the PI probe does not produce convincing labeling (more on that below). Therefore, the abstract should be amended, perhaps to say something more diplomatic, like "detection of all phosphoinositide species for which suitable probes exist". I was personally bummed there was not a good probe for PI5P. This is not the authors fault, but I had a niggling hope given their recent discovery of the long elusive PI(3,5)P2 probe!

We have amended the abstract and have taken out the claim to detect all 8 phosphoinositides. I (H. Maib) am at least as bummed as the reviewer about the lack of a good PI(5)P probe. Indeed, I have undertaken considerable effort to identify a potential probe but have thus far failed. The levels of PI(5)P are just so far lower than that of any of the other phosphoinositides that any reliable probe would need to be incredibly specific to be useful in cells. Thus far, all of the effectors I have looked at also bind to other PIPs to some degree, making them useless. I have not given up hope yet but if I should ever be successful in finding a reliable PI(5)P probe, it will be a paper in its own rights and will shed some much needed light on this most elusive phosphoinositide. My current best guess is to look into the family of MTMRs, but I wouldn't hold my breath...

The PI probe: the diffuse staining pattern is inconclusive. This could be caused by the ubiquity of the lipid in intracellular membranes as the authors suggest, but it could also be non-selective. I have two suggestions that could shed light on which it is: The most trivial is that Pemberton et al (and Zewe et al) reported ubiquitous, widespread PI distribution - but not in the plasma membrane. So, this might be conveniently shown by multiplexed imaging with the PH-PLCD1 probe. Secondly, some demonstration that the labelling is PI-dependent might help. One idea is to use a bacterial PI-PLC protein (the Bacillus protein can be bought from Sigma) in the fixed cells prior to staining to deplete the PI, maybe during the blocking step. Ice-cold temperatures may inhibit the enzyme, though I have had good experience with other bacterial enzymes working on ice so I think it is worth a try.

The PI staining was perhaps the most difficult to address. We have reasoned that by blocking all lipid kinases that use PI as substrate, we should see an increased level of this phosphoinositide on some membranes. Indeed, after treating HeLa cells with an inhibitor cocktail consisting of Wortmannin, PI-273, GSK-A1, PI4KIII β -IN 10 and Apilimod (at 500nM each for 30min) and multiplex staining against PI, PI(4)P and PI(3)P, we are able to detect an increased signal of PI on the membranes of some vacuolar structures, but the labelling still shows a lot of diffuse background. While we believe that the visualisation of PI is possible using this probe, it requires careful controls. We have added these caveats to the discussion. We also see no PI on the PM as suggested. Interestingly, we have included the staining against PI(4)P and PI(3)P to confirm their depletion and while PI(3)P is indeed completely depleted from endo/lysosomes, some remnants of PI(4)P still remain, indicating that PI(4)P might have a longer turnover period than PI(3)P. We have included this data as supplemental Figure S2a:

Figure S2: a) HeLa cells were treated with a kinase inhibitor cocktail of Wortmannin, PI-273, GSK-A1, PI4KIII β -IN 10 and Apilimod each at 500nM for 30min before being fixed and stained using recombinant biosensors against PI, PI(4)P and PI(3)P conjugated to Alexa488, 546 or 647 respectively. Linescan shows signal of PI on the membrane of large vacuoles that can be found throughout the cell.

The PI3P probe: The staining looks very convincing, but to support the statement of "clear labelling of endo/lysosomal structures throughout the cell", co-localization with markers should be used. This may even be doable after the post-fixation step to stain for markers such as EEA1 or LAMP1. Alternatively, the authors could use expressed markers such as Rabs, or even endogenously tagged proteins where available (as they did for Arf1). Perhaps the simplest approach would be to show co-localization with the expressed EGFP-FYVE-HrsX2!

The PI4P probe: As for PI3P, some co-localization should be provided to substantiate the claim of localization to "trafficking vesicles". Late endocytic and recycling compartments are the two that have been reported in the literature. The Golgi labelling is substantiated in Figure 4. The plasma membrane staining is also somewhat hard to make out (compared to GFP-SidC when expressed). Are there clearer examples?

Using a pulse chase assay of fluorescently labelled transferrin we address the nature of the PI(3)P or PI(4)P containing endosomal compartments at the same time. This elegantly shows that PI(3)P positive endosomes are mainly early endocytic structures while PI(4)P are indeed late endocytic and recycling compartments. We have included this data as Figure 4a:

Figure 4: a) Pulse-chase assay in HeLa cells using fluorescently labelled transferrin followed by staining against PI(3)P and PI(4)P using recombinant biosensors. Signal overlap was determined using the ImageJ plugin Squassh. Data is mean \pm SEM from three independent repeats with 16-20 cells each

In addition, selectivity of the pools can be demonstrated by elimination of specific pools using selective PI4K inhibitors, which are commercially available. These include PM PI4KIIIalpha (GSK-A1, Sigma), PI-273 for endosomal PI4KIIalpha and PI4KIIIbeta-IN-10 for Golgi PI4KIIIbeta (both MedChemExpress).

We did exactly as the reviewer suggested and treated HeLa cells with the inhibitors at 100nM for 30min. Treatment with GSK-A1 depleted PI(4)P from the PM. Treatment with PI4KIIIβ-IN 10 had the strongest effect on PI(4)P localisation and depleted Golgi pools and to some extent the PM, with some endosomal pools remaining. PI-273 had the weakest effect on PI(4)P localisation but we noted a decreased endosomal localisation. All in all, this experiment actually worked as the reviewer has predicted and is part of Supplemental Figure S2b:

Figure S2 b) HeLa cells were treated with either GSK-A1, PI-273 or PI4KIIIβ-IN 10 at 100nM each for 30min before being fixed and stained against PI(4)P using recombinant biosensors against PI(4)P conjugated to Alexa488

PIP3 and PI(3,4)P2 probes: The majority of these lipids are synthesized by class I PI3K, for which there are many commercially available, potent inhibitors (e.g. GDC-0941). Depletion of the staining by these compounds should be shown as a minimum standard for specificity in cells. Furthermore, given that these lipids are predominantly formed during receptor signaling events, enhanced production following cellular stimulation with e.g. EGF or insulin in HeLa cells would enhance the confidence in the probes.

To address the PI(3,4,5)P₃ and PI(3,4)P₂ staining, we have used EGF stimulation to induce membrane ruffles in which these phosphoinositides are enriched. To test the specificity of our staining, we treated HeLa cells with the PI3K inhibitor LY294002 (1μm for 30min) prior to EGF stimulation (5ng/ml for 10min) and multiplex stained against PI(4,5)P₂ PI(3,4)P₂ and PI(3,4,5)P₃. We find that while membrane ruffles still form, they are heavily depleted of both PI(3,4,5)P₃ and PI(3,4)P₂ while not affecting PI(4,5)P₂. Also note that LY treatment does not affect the background or nuclear staining observed with the PI(3,4,5)P₃ probe. We included this data in Supplemental figure S2c:

Figure S2c) HeLa cells were treated with LY 294022 at 1μM for 30min, stimulated with 5ng/ml human EGF for 10min and fixed and stained using recombinant biosensors against PI(4,5)P₂, PI(3,4)P₂ and PI(3,4,5)P₃ conjugated to Alexa488, 546 or 647 respectively. Line-scans show signal of PI(4,5)P₂, PI(3,4)P₂ and PI(3,4,5)P₃ in DMSO and LY294002 treated cells at membrane ruffles as well as the cytosolic background.

All probes: Inclusion of millimolar concentrations of neomycin in the blocking and staining solutions will most likely block all the polyphosphoinositides, and be another nice specificity control. I wouldn't be surprised if it also worked for the PI probe too! This would be a very easy control experiment.

This is a great suggestion; however we believe that the suggested treatment with kinase inhibitors has been very successful in controlling for the specificities of the stainings.

The manuscript states in the results that: "PI(3,5)P₂ on contrast is located on the same membrane as PI(3)P". As I read this, I was struck by the notion that every cartoon in every review I have read on this topic has PI3P and PI(3,5)P₂ defining distinct membrane compartments in the cell (normally, early vs late endocytic compartments). Given the substrate for PIKfyve, of course they would co-localize. But as the first demonstration of this fact, I think the authors could emphasize this point more, perhaps in the discussion on the ramifications for defining late vs early endosomal identity with these lipids.

We are delighted to hear about this and hope that the availability of the PI(3,5)P₂ probe will alter the way that we perceive the distribution and interrelationship of PI(3)P and PI(3,5)P₂. While not necessarily the focus of this manuscript, we are currently undertaking multiple lines of enquiry into the conversion and subcellular distribution of PI(3)P → PI(3,5)P₂ and hope to publish more work focused on this in the not-too-distant future. I am now also considering writing a short review article with Jason King about the PI(3)P → PI(3,5)P₂ conversion and where these lipids localise.

Value as a tool/method: This protocol can indeed be easily implemented by other labs, justifying its publication in a high profile journal. Distribution of the constructs in Addgene is a requisite for this, and I commend the authors for doing this. An additional suggestion (though not a pre-requisite for publication) is to see if a company is interested in producing the proteins for distribution commercially? Echelon Biosciences do sell some GST-tagged probes and may be interested in providing this as a service with the probes reported (assuming there are no licensing issues with the use of SNAP, for example). Commercial availability would, I suspect, greatly enhance adoption of the approach.

We have made all of the constructs (expect the dodgy PI(5)P probe) available on Addgene and hope for a wide adoption of our approach. Commercialisation is difficult due to the patent on the SNAP tag technology by NEB.

Technical comments

Figure 2: the number of beads quantified is stated, but this should be supported by the number of independent experiments that these were quantified from.

Done

The source and validity of the cell lines used should be indicated in the methods.

The cell lines have been kind gifts from my colleagues at the University of Sheffield and this has been added to the manuscript.

The composition of the PIPES buffer is not stated. This could be important!

Done

What is the source of the saponin used? This is a reagent that can be quite variable given the source, so stating the manufacturer and catalogue number will maximize reproducibility.

The Saponin used was from Sigma S7900 from quillaja bark. This has been added to the methods section.

Reviewer #3 (Comments to the Authors (Required)):

The authors of this study generated plasmids that enable expression, purification and fluorescent labeling of biosensors for phosphoinositides. Fluorescent protein tagged biosensors have been reported and the use of a SNAP tag increases flexibility in the use of fluorescent labels. The biosensors were characterised in vitro and subsequently used to image phosphoinositides at high resolution in fixed cells, spheroids and drosophila pupal wings.

My main worry is that it is unclear how the lipids are fixed by crosslinking with formaldehyde. Next to this (although the selectivity is nicely tested in an in vitro system) it is not clear to what extent the biosensors exhibit aspecific binding to cellular components. Some evidence of aspecific binding seems to present in the images in figure 3, i.e. labeling of nucleoli for several of the probes and the reticular labeling observed with the PtdIns(3,4,5)P3 probe seems aspecific as well. Below I will list my comments that hopefully can be used to improve this work.

We thank the reviewer for the time and careful revision of our work and hope to have addressed most of their concerns in the revised version.

The fixation method using 4% PFA and 0.2% GA will highly crosslink amine groups but is indeed unlikely to fix lipids. Therefore, the phosphoinositides that we are able to detect are most likely freely available at the time of fixation. Using ectopic overexpression of a fluorescently labelled biosensor we can indeed demonstrate that high expression levels almost completely block detection by recombinant biosensors, while low expression levels still permit their detection. Thus, our approach is sensitive to the free population of phosphoinositides that are present by the time of fixation. Minor pools of phosphoinositides might therefore escape detection by biosensors due to being highly occupied by endogenous effectors. This data is now presented in figure S1. And we have addressed these concerns in the discussion.

We have further undertaken extensive controls using lipid kinase inhibitors to control for the specificity of our stainings. This is shown in figure S2. Importantly we never see any changes in the nuclear staining observed with most of these probes, even after kinase inhibition. We strongly believe the nuclear staining pattern to be unspecific, due to the strong negative charge and high availability of phosphate groups in open DNA and have specified this in the discussion section.

Please find below a detail response to the individual points raised.

1. In the abstract "we present a toolkit for the reliable, fast, multiplex, and super-resolution detection of all 8 phosphoinositides" the use of "multiplex" here suggests (at least, this is what I understood before reading the entire paper) that it is possible to image all 8 at the same time. I recommend to rephrase the sentence to avoid this suggestion.

We have rephrased the abstract and hope to have made it more clear that we are not able to detect all 8 phosphoinositide species.

2. The authors state that "liposomes composed of 95% 1- palmitoyl-2-oleoyl-glycero-3-phosphocholine (POPC), and 5% of each phosphoinositide. These membrane coated beads are an ideal substrate to evaluate binding specificities, as they faithfully mimic the charge and fluidity of biological membranes (Pucadyil and Schmid, 2010)".

This statement raises several questions:

-The statement suggests that the coated beads mimic fluidity. In the study that is cited, DOPC/DOPS and DOPE were used, which have different acyl chains, and therefore a different fluidity. So, I'm not sure why this reference is used here and there might be better citations to support the idea that these beads "mimic the charge and fluidity of biological membranes".

The work from Pucadyil and Schmid, 2010 was seminal for the use of silica beads as templates for supported lipid bilayer generation. The use of these beads allows for the reproducible and straight forward production of supported lipid bilayers and each bead can be used to quantify the recruitment of fluorescently labelled biosensors. This was crucial to enable the extensive quantification of effector binding performed in this manuscript (in total 61.440 beads have been imaged in Figure 2). The fluidity of the membranes that we have used here might indeed be different from those in Pucadyil and Schmid, 2010 and we have taken out that claim.

-I would like to see citations to support the notion that 5% of phosphoinositides is a decent percentage. It's my understanding that PtdIns levels are much higher than PtdIns(3,4,5)P₃. What at natural levels and how to these compare to the 5% used here?

The reviewer is very right that the levels of phosphoinositides in cells is highly variable and that while PI represents 10-20% of total cellular phospholipids, P(4)P and PI(4,5)P₂ constitute only ~0.2–1%. The intention for the membrane coated beads was never to represent the natural occurring quantities of phosphoinositides but to test the specificities of the recombinant probes towards them. This requires the same amount of each of the phosphoinositides in the membranes to directly compare effector binding between them. For this purpose, 5% is a commonly used amount in reconstitution of effector binding as in these studies: Fracchiolla D et al J Cell Biol. 2020. Tan and Finkel, Nature, 2022. Murray et al Nature, 2016. Busse et al. Autophagy. 2013

-Since the charge increases with the number of phosphates on the inositol ring, there's a difference in charge between the PIs, most prominently between PtdIns(3,4,5)P₃ and PI. Do they authors account for this in the in vitro assay? Would it make sense to include a control with POPC + POPA or POPS to study the effect of charge on binding of the biosensor?

In our assay, the increase in negative charge of inositol polyphosphates serves as control to study the effect on charge-based binding in itself. As the reviewer rightly points out, the charge will increase from PI → PI(3,4,5)P₃, therefore a charge based binding effect would manifest itself as stepwise increase in binding between the mono- and di-phosphates. This is especially useful for the unspecific binding observed for the PI(5)P probe as we have pointed out: "...Even though this probe shows stronger binding to PI(5)P compared to PI, PI(3)P, PI(3,4)P₂ and PI(4)P, it binds even more strongly to the other 5' containing di- and tri-phosphates, suggesting a charge based binding effect in addition to 5' binding (Fig. 2d)....". While inclusion of high amounts of POPS might be useful to model the internal leaflet of the PM, most other membranes do not usually contain high amounts of negative charge and we therefore find the minimal approach of 95% POPC + 5% phosphoinositides to be the most representative way to compare the specificities of each biosensor toward the family of phosphoinositides.

3. From what I read, the His tag is still present. This may not be a problem, but did the authors study the effect of the His tag? Does the His tag affect specificity and may it explain some of the binding in the nucleus, e.g. for the PtdIns(4,5)P₂ sensor?

We have done as suggested by the reviewer and cleaved the 6xHis tag of the PI(4,5)P₂ probe by addition of PreScission protease and stained HeLa cells as previously. From what we can see, the staining is indistinguishable from stainings still containing the His tag. The nuclear staining pattern is therefore unlikely to be a result of the His tag but more likely due to the negative charge and high concentration of phosphate groups in open DNA. This is now included in supplemental Figure S3.

Figure S3: Staining using one step his purification. The Recombinant biosensor against PI(4,5)P₂ (6xHis-SNAP-PLCδ1) was purified from BL21 e.Coli using a HisTrap column and eluted against an increasing gradient of 250mM Imidazole. Peak fractions were pooled and either directly labelled with SNAP-Surface Alexa488, or first incubated with recombinant PreScission Protease, to cleave of the 6xHis tag and used to stain HeLa cells as previously described.

4. The *in vitro* reconstitution and binding assay is nice, as it allows to directly observe the binding. In the data of figure 2 it is not clear whether these are from a single experiment, or from multiple replicates? In case of the second scenario, it would be more transparent to show the data as superplot.

The data is pooled from three technical repeats. This has been added to the figure legend. Indeed, each separate bead can be considered as technical replicate as it represents a separate binding reaction. As there is no real “biological” replicate in a pure *in vitro* system we feel that displaying the median with interquartile range of all beads is the most transparent way to visualise the data.

5. Does the intensity presented in figure 2 reflect binding efficiency (I think so), and would it in that case be sensible to define a threshold for background binding? For instance, it seems like a value of 500 and anything below is background and the authors may consider showing this threshold as a line in the plots.

We feel that the amount of “background” binding to the membrane is an important property of the probes and represents binding that is independent of the phosphoinositide composition of the membrane. As can be seen by the PI(3)P probe that has the lowest background binding at ~120 a.u. while the PI(4,5)P₂ probe has a background binding of ~350 a.u. while both probes bind to their respective phosphoinositide with very similar intensities (1760 vs 1806

a.u). We therefore feel that the most honest way to represent the data is as raw as possible and that by denoting some binding as “background” we would obscure from the fact the probes also have a varying, unspecific binding component to the membranes.

6. Related to the previous point, did the authors think of a control to determine background or aspecific binding in this assay?

We are of the opinion that the background and aspecific binding can directly be inferred from the data in figure 2. This is apparent in the binding to membranes containing the separate phosphoinositides and the most transparent way is to directly present the data as raw as possible. Importantly all membranes containing mono-, or di- phosphoinositides will have the same charge and are therefore directly comparable to determine unspecific binding.

7. In figure 3, DAPI seems to be included, but I do not see that mentioned in the legend or material and methods?

Thank you for pointing out this oversight. This has been added to the legend.

8. Several probes show localization in the nucleoli (if I interpret the images correctly). This should be clearly highlighted, because in the current manuscript it is only mentioned for PtdIns(5)P.

We have included a section about the nucleolar staining patterns in the discussion. We agree with the reviewer that it is important to highland this concern and write that:

“...Finally, we caution against the over interpretation of the nuclear staining using most of these biosensors. The probes against PI, PI(3)P, PI(5)P, PI(3,5)P₂, PI(4,5)P₂ and PI(3,4,5)P₃ all show nuclear staining patterns, and we believe that this is most likely unspecific due to the negative charge and the high availability of phosphate groups in open DNA. Importantly, the probes against PI(3)P, PI(3,5)P₂ and PI(3,4,5)P₃ show strong nuclear localisation which is completely unaffected by treatment with well characterised lipid kinase inhibitors. While the role of nuclear phosphoinositides is an important field of research (Shah et al., 2013), it is unlikely that this staining approach is able to reliably detect them.”

9. In figure 5C, the biosensors are used in Drosophila pupal wings. The PtdIns(4)P probe shows diffuse staining and it discussed that "Noticeably, PI(4)P is also enriched at these junctions albeit with a more diffuse staining pattern, indicating the presence of intracellular PI(4)P membranes in close proximity to the plasma membrane (Fig. 5c). These observations are in good agreement with the dynamic regulation of these two lipids at cell junctions in Drosophila tissue to drive recruitment of polarity proteins through electrostatic interactions (Dong et al., 2015; Lu et al., 2022)"

When I look at the paper by Lu et al (<https://elifesciences.org/articles/79582>), the localization of PtdIns(4)P with the SidM probe in drosophila is at the plasma membrane, with very little diffuse, intracellular labeling. This seems to be quite different from the diffuse staining that the author report here (figure 5C). Is the diffuse staining real, i.e. is it specific labeling of PtdIns(4)P? This needs to be discussed and compared to data from live cell imaging.

The reviewer is right that the localisation of PI(4)P in Lu et al seems to somewhat differ to the staining against PI(4)P with the recombinant biosensors in our hands. This can be due to many reasons. Among them is that the ectopically expressed SidM probe might recognise PI(4)P at the PM more robustly than at intracellular compartments. The presence of intracellular pools of PI(4)P is very well known, therefore the question should rather be “why is there no intracellular labelling in Lu et al?”. Further, we do not know what Z-Position the images in Lu et al were acquired at. As seen in the figure below, when imaged at the very top of the cells, there is less intracellular PI(4)P visible in our staining approach. It is therefore difficult to directly compare our images to those of Lu et al, since they might be taken at a different Z position and have used a different biosensor.

PI(4)P staining at different z-positions *Drosophila* pupal wings.

10. The authors generate biosensors with different fluorophores (Alexa488, 546 and 647) for multiplex imaging. Is anything known about the labeling efficiency and differences between the probes in terms of efficiency? In addition to using this for labeling on fixed cells, these probes could be used to strengthen the in vitro characterisation. For instance, a competition experiment between two similar binding domains with different labels. Or co-labeling of beads with two biosensors for different phosphoinositides, to show that two different lipids can indeed be detected on the same bead.

We have not investigated different labelling efficiencies of the SNAP ligands and feel that this is outside of the scope of the manuscript. Which is focused on the use of these probes for the visualisation of phosphoinositides in a biological context. More *in vitro* characterisations of the probes might be a valuable undertaking for work that is more focused on their use for *in vitro* reconstitution approaches.

11. Related to the previous point, if a red biosensor (or unlabeled) effectively competes with a green biosensor on the beads (and that's expected), I would think that a similar strategy can be used in cells to verify specificity of the signal. For instance green and red biosensor can be mixed in a 1:9 ratio and in this situation the green channel would predominantly show the aspecific labeling.

Competition experiments using differently labelled biosensors are a valuable tool to characterise the affinities of different biosensors for the same phosphoinositide (eg fluorescently labelled PH-Tuby vs PLC δ 1). However, we are able to draw on decades of experience by multiple groups that have already quantified the affinities of these well-known biosensors previously (reviewed in detail in Wills et al. Mol Biol Cell, 2018). We further feel that the data presented in figure 2 comprehensively allows for direct comparison of the specificities and background binding of each biosensor *in vitro*. Importantly, this manuscript

explores their use for cellular staining approaches and the specificities of the stainings in that context have now been addressed using a range of lipid kinase inhibitors.

12. In the end, this would be a toolkit that should be of use to others. This requires more detail in the materials and methods. For instance, what was the method for the e.coli lysis, what was the buffer for anion exchange and the size exclusion column, what is the composition and pH of "PIPES buffer"? It would be informative to state the volume of e.coli culture that was used for the isolation and what the typical yields were obtained? How much protein was labeled in what volume and for how many experiments can that be used (approximately).

We have added these details in the materials and methods section. Ultimately, we do not feel it helpful to provide a detailed step by step guide for the purification of these biosensors since the purification of His tagged proteins from BL21 is a standard practise in most labs at this point. There are many different ways to purify recombinant proteins via a His tag and we do not want to give the false impression of any method being inherently superior over the other. We have added more detail about how we have performed the purification procedure but also want to give people the freedom to perform the purification using their own preferred methods, which might deviate from ours.

13. Did the authors check whether a one step isolation with His tag is sufficient, as this would greatly improve the user-friendliness of the method?

We have done as the reviewer suggested and purified the recombinant PI(4,5)P₂ probe via a one-step his tag by elution against a gradient of 250mM Imidazole and labelled the peak fractions using SNAP-Surface Alexa488. Since the SNAP-ligand will only react with the SNAP tagged biosensor, this crude purification procedure could be useful for some users. This has been included in Figure S3 and we state that: "...Finally, by using a one-step His purification followed by removing the tag using PreScission protease, we verify that the presence of the 6xHis tag does not affect the staining and enables the use of a simplified one-step purification, if so desired (Supplemental Figure 3)."

Reviewed by Joachim Goedhart (University of Amsterdam).

March 6, 2024

RE: JCB Manuscript #202310095R

Dr. Hannes Maib
University of Sheffield
Biosciences
Western Bank
Sheffield S10 2TN
United Kingdom

Dear Dr. Maib:

Thank you for submitting your revised manuscript entitled "Recombinant biosensors for multiplex and super-resolution imaging of phosphoinositides". We would be happy to publish your paper in JCB pending final revisions necessary to meet our formatting guidelines (see details below).

A. MANUSCRIPT ORGANIZATION AND FORMATTING:

- 1) Text limits: Character count for Articles is < 40,000, not including spaces. Count includes abstract, introduction, results, discussion, and acknowledgments. Count does not include title page, figure legends, materials and methods, references, tables, or supplemental legends.
- 2) Figures limits: Articles may have up to 10 main text figures.
- 3) Figure formatting: Scale bars must be present on all microscopy images, including inset magnifications. Molecular weight or nucleic acid size markers must be included on all gel electrophoresis.
- 4) Statistical analysis: Error bars on graphic representations of numerical data must be clearly described in the figure legend. The number of independent data points (n) represented in a graph must be indicated in the legend. Statistical methods should be explained in full in the materials and methods. For figures presenting pooled data the statistical measure should be defined in the figure legends. Please also be sure to indicate the statistical tests used in each of your experiments (either in the figure legend itself or in a separate methods section) as well as the parameters of the test (for example, if you ran a t-test, please indicate if it was one- or two-sided, etc.). Also, if you used parametric tests, please indicate if the data distribution was tested for normality (and if so, how). If not, you must state something to the effect that "Data distribution was assumed to be normal but this was not formally tested."
- 5) Abstract and title: The abstract should be no longer than 160 words and should communicate the significance of the paper for a general audience. The title should be less than 100 characters including spaces. Make the title concise but accessible to a general readership.

* We suggest the following edits to your title and running title *

Title: Recombinant biosensors for multiplex and super-resolution phosphoinositide imaging

Running title: Recombinant biosensors for phosphoinositide imaging

6) Materials and methods: Should be comprehensive and not simply reference a previous publication for details on how an experiment was performed. Please provide full descriptions in the text for readers who may not have access to referenced manuscripts.

7) * All antibodies, cell lines, animals, and tools used in the manuscript should be described in full, including accession numbers for materials available in a public repository such as the Resource Identification Portal. Please be sure to provide the sequences for all of your primers/oligos and RNAi constructs in the materials and methods. You must also indicate in the methods the source, species, and catalog numbers (where appropriate) for all of your antibodies. Please also indicate the acquisition and quantification methods for immunoblotting/western blots.*

8) Microscope image acquisition: The following information must be provided about the acquisition and processing of images:

- Make and model of microscope
- Type, magnification, and numerical aperture of the objective lenses
- Temperature
- Imaging medium
- Fluorochromes
- Camera make and model
- Acquisition software
- Any software used for image processing subsequent to data acquisition. Please include details and types of operations involved (e.g., type of deconvolution, 3D reconstitutions, surface or volume rendering, gamma adjustments, etc.).

10) Supplemental materials: There are strict limits on the allowable amount of supplemental data. Articles may have up to 5 supplemental figures. Please also note that tables, like figures, should be provided as individual, editable files. A summary of all supplemental material should appear at the end of the Materials and methods section.

13) ORCID IDs: ORCID IDs are unique identifiers allowing researchers to create a record of their various scholarly contributions in a single place. Please note that ORCID IDs are now *required* for all authors. At resubmission of your final files, please be sure to provide your ORCID ID and those of all co-authors.

Please note that JCB now requires authors to submit Source Data used to generate figures containing gels and Western blots with all revised manuscripts. This Source Data consists of fully uncropped and unprocessed images for each gel/blot displayed in the main and supplemental figures. Since your paper includes cropped gel and/or blot images, please be sure to provide one Source Data file for each figure that contains gels and/or blots along with your revised manuscript files. File names for Source Data figures should be alphanumeric without any spaces or special characters (i.e., SourceDataF#, where F# refers to the associated main figure number or SourceDataFS# for those associated with Supplementary figures). The lanes of the gels/blots should be labeled as they are in the associated figure, the place where cropping was applied should be marked (with a box), and molecular weight/size standards should be labeled wherever possible.

Journal of Cell Biology now requires a data availability statement for all research article submissions. These statements will be published in the article directly above the Acknowledgments. The statement should address all data underlying the research presented in the manuscript. Please visit the JCB instructions for authors for guidelines and examples of statements at (<https://rupress.org/jcb/pages/editorial-policies#data-availability-statement>).

B. FINAL FILES:

Thank you for your attention to these final processing requirements. Please revise and format the manuscript and upload materials within 7 days. If you need an extension for whatever reason, please let us know and we can work with you to determine a suitable revision period.

Thank you for this interesting contribution, we look forward to publishing your paper in Journal of Cell Biology.

Sincerely,

Tamas Balla, MD, PhD
Monitoring Editor

Andrea L. Marat, PhD
Senior Scientific Editor

Journal of Cell Biology

Reviewer #1 (Comments to the Authors (Required)):

In this revised version of their manuscript, Maib and coworkers have addressed all my concerns. I appreciate the thorough revision work and the additional controls performed.

I look forward to see this new generation of phosphoinositide probes used by the community, and congratulate the authors on this nice and important work.

Reviewed by Kay Oliver Schink (University of Oslo)

Reviewer #2 (Comments to the Authors (Required)):

The authors have comprehensively addressed all three reviewer's comments. They have re-written the manuscript or added elegant new experiments that addressed all of my concerns. Therefore, I think the paper is ready for publication!

Reviewer #3 (Comments to the Authors (Required)):

I thank the authors for the detailed answers to all of the questions and the new data that is included. This study will add a valuable set of tools to study phosphoinositides and I'd like to congratulate the authors with this achievement.

(Joachim Goedhart, University of Amsterdam, NL).